# Understanding Deep Gradient Leakage via Inversion Influence Functions

**Haobo Zhang**[*]
Michigan State University
zhan2060@msu.edu

**Junyuan Hong**[*]
Michigan State University
University of Texas at Austin
jyhong@utexas.edu

**Yuyang Deng**
Pennsylvania State University
yzd82@psu.edu

**Mehrdad Mahdavi**
Pennsylvania State University
mahdavi@cse.psu.edu

**Jiayu Zhou**
Michigan State University
jiayuz@msu.edu

## Abstract

Deep Gradient Leakage (DGL) is a highly effective attack that recovers private training images from gradient vectors. This attack casts significant privacy challenges on distributed learning from clients with sensitive data, where clients are required to share gradients. Defending against such attacks requires but lacks an understanding of *when and how privacy leakage happens*, mostly because of the black-box nature of deep networks. In this paper, we propose a novel Inversion Influence Function ($I^2F$) that establishes a closed-form connection between the recovered images and the private gradients by implicitly solving the DGL problem. Compared to directly solving DGL, $I^2F$ is scalable for analyzing deep networks, requiring only oracle access to gradients and Jacobian-vector products. We empirically demonstrate that $I^2F$ effectively approximated the DGL generally on different model architectures, datasets, modalities, attack implementations, and perturbation-based defenses. With this novel tool, we provide insights into effective gradient perturbation directions, the unfairness of privacy protection, and privacy-preferred model initialization. Our codes are provided in https://github.com/illidanlab/inversion-influence-function.

## 1 Introduction

With the growing demands for more data and distributed computation resources, distributed learning gains its popularity in training large models from massive and distributed data sources (McMahan et al., 2017; Gemulla et al., 2011). While most distributed learning requires participating clients to share only gradients or equivalents and provides data confidentiality, adversaries can exploit such information to infer training data. Deep Gradient Leakage (DGL) showed such a real-world risk in distributed learning of deep networks (Zhu et al., 2019; Jin et al., 2021). Only given a private gradient and the corresponding model parameters, DGL reverse-engineers the gradient and recovers the private images with high quality. The astonishing success of the DGL started an intensive race between privacy defenders (Sun et al., 2020; Wei et al., 2021; Gao et al., 2021; Scheliga et al., 2022) and attackers (Geiping et al., 2020; Zhao et al., 2020; Jin et al., 2021; Jeon et al., 2021; Zhu and Blaschko, 2021; Fowl et al., 2022).

A crucial question to be answered behind the adversarial game is *when and how DGL happens*, understanding of which is fundamental for designing attacks and defenses. Recent advances in

---

[*]Equal contribution.

37th Conference on Neural Information Processing Systems (NeurIPS 2023).

theoretical analysis shed light on the question: Balunović et al. (2022) unified different attacks into a Bayesian framework, leading to Bayesian optimal adversaries, showing that finding the optimal solution for the DGL problem might be non-trivial and heavily relies on prior knowledge. Pan et al. (2022); Wang et al. (2023b) advanced the understanding of the leakage in a specific class of network architectures, for example, full-connected networks. However, it remains unclear how the privacy leakage happens for a broader range of models, for example, the deep convolutional networks, and there is a pressing need for a *model-agnostic* analysis.

In this paper, we answer this question by tracing the recovered sample by DGL algorithms back to the private gradient, where the privacy leakage ultimately derives from. To formalize the optimal DGL adversary without practical difficulties in optimization, we ask the counterpart question: *what would happen if we slightly change the private gradient?*

Answering this question by perturbing the gradient and re-evaluating the attack can be prohibitive due to the difficulty of converging to the optimal attack in a highly non-convex space. Accordingly, we propose a novel Inversion Influence Function ($I^2F$) that provides an analytical description of the connection between gradient inversion and perturbation. Compared to directly solving DGL, the analytical solution in $I^2F$ efficiently scales up for deep networks and requires only oracle access to gradients and Jacobian-vector products. We note that $I^2F$ shares the same spirit of the influence function (Koh and Liang, 2017; Hampel, 1974) that describes how optimal model parameters would change upon the perturbation of a sample and $I^2F$ can be considered as an extension of the influence function from model training to gradient inversion.

The proposed $I^2F$ characterizes leakage through the lens of private gradients and provides a powerful tool and new perspectives to inspect the privacy leakage: (1) First, we find that gradient perturbation is not homogeneous in protecting the privacy and is more effective if it aligns with the Jacobian singular vector with smaller singular values. (2) As the Jacobian hinges on the samples, the variety of their Jacobian structures may result in unfair privacy protection under homogeneous Gaussian noise. (3) We also examine how the initialization of model parameters reshapes the Jocabian and therefore leads to quite distinct privacy risks.

These new insights provide useful tips on how to defend DGL, such as perturbing gradient in specific directions rather than homogeneously, watching the unfairness of protection, and carefully initializing models. We envision such insights could lead to the development of fine-grained privacy protection mechanisms. Overall, our contributions can be summarized as follows. (1) For the first time, we introduce the influence function for analyzing DGL; (2) We show both theoretical and empirical evidence of the effectiveness of the proposed $I^2F$ which efficiently approximates the DGL in different settings; (3) The tool brings in multiple new insights into when and how privacy leakage happens and provides a tool to gauge improved design of attack and defense in the future.

## 2 Related Work

*Attack.* Deep Gradient Leakage (DGL) is the first practical privacy attack on deep networks (Zhu et al., 2019) by only matching gradients of private and synthetic samples. As accurately optimizing the objective could be non-trivial for nonlinear networks, a line of work has been developed to strengthen the attack (Zhao et al., 2020). Geiping et al. (2020) introduce scaling invariance via cosine similarity. Balunović et al. (2022) summarizes these attacks as a variety of prior choices. The theorem is echoed by empirical studies that show better attacks using advanced image priors (Jeon et al., 2021). It was shown that allowing architecture modification could weigh in higher risks in specific cases (Zhu and Blaschko, 2021; Fowl et al., 2022). Besides the study of gradient leakage in initialized models, recent evidence showed that the well-trained deep network can also leak the private samples (Haim et al., 2022). Yet, such an attack still requires strict conditions to succeed, e.g., small dataset and gradient flow assumptions. Though these attacks were shown to be promising, exactly solving the attack problem for all types of models or data is still time-consuming and can be intractable in reasonable time (Huang et al., 2021).

*Defense.* Most defense mechanisms introduce a perturbation in the pipeline of gradient computation. *1) Gradient Noise.* Since the DGL is centered on reverse engineering gradients, the straightforward defense is to modify the gradient itself. Zhu et al. (2019) investigated a simple defense by Gaussian noising gradients or pruning low-magnitude parameters-based defense. The Gaussian noise was motivated by Differential Privacy Gradient Descent (Dwork, 2006; Abadi et al., 2016), which

guarantees privacy in an aggregated gradient. Except for random perturbation, the perturbation could also be guided by representation sensitivity (Sun et al., 2020) or information bottleneck Scheliga et al. (2022). *2) Sample Noise.* The intuition of sample noise is to change the source of the private gradient. For example, Gao et al. (2021) proposed to seek transformations such that the transformed images are hard to be recovered. Without extensive searching, simply using strong random data augmentation, e.g., MixUp (Zhang et al., 2017), can also help defense (Huang et al., 2021). If public data are available, hiding private information inside public samples is also an alternative solution (Huang et al., 2020). Though the defenses showed considerate protection against attacks in many empirical studies, the security still conceptually relied on the expected risks of a non-trivial number of evaluations. Explanation of how well a single-shot protection, for example, noising the gradient, is essential yet remains unclear.

*Understanding DGL.* There are many recent efforts in understanding when noise-based mechanisms help privacy protection. Though Differential Privacy (DP) provides a theoretical guarantee for general privacy, it describes the worst-case protection and does not utilize the behavior of DGL, motivating a line of studies to understand when and how the DGL happens. Sun et al. (2020) connected the privacy leakage to the representation, which aligns with later work on the importance of representation statistics in batch-normalization layers (Hatamizadeh et al., 2021; Huang et al., 2021). Balunović et al. (2022) unified different attacks in a Bayesian framework where existing attacks differ mainly on the prior probability. Chen and Campbell (2021) investigated the model structure to identify the key reasons for gradient leakage. Hatamizadeh et al. (2021); Huang et al. (2021) evaluated the DGL in practical settings and showed that BN statistics is critical for inversion. There are efforts advancing theoretical understandings of DGL. Pan et al. (2022) studied the security boundary of ReLU activation functions. In a specific class of fully-connected networks, Wang et al. (2023b) showed that a single gradient query at randomized model parameters like DP protection can be used to reconstruct the private sample. Hayes et al. (2023); Fan et al. (2020) proposed an upper bound of the success probability of the data reconstruction attack and the relative recovery error, respectively, which is considered as the least privacy risk. Instead, we consider the worst-case privacy risk.

Though many useful insights are provided, existing approaches typically focus on specific model architectures. This work provides a new tool for analyzing the DGL without assumptions about the model architectures or attacking optimization. Independent of the model architectures, our main observation is that the Jacobian, the joint derivative of loss w.r.t. input and parameters, is the crux.

## 3    Inversion Influence Function

In this section, we propose a new tool to unravel the black box of DGL. We consider a general loss function $L(x, \theta)$ defined on the joint space of data $\mathcal{X}$ and parameter $\Theta$. For simplicity, we assume $x$ is an image with oracle supervision if considering supervised learning. Let $x_0$ be the private data and $g_0 \triangleq \nabla_\theta L(x_0, \theta)$ be its corresponding gradient which is treated as a constant. An inversion attack aims at reverse engineering $g_0$ to obtain $x_0$. Deep Gradient Leakage (DGL) attacks by directly searching for a synthetic sample $x$ to reproduce the private gradient $g$ by $\nabla_\theta L(x, \theta)$ (Zhu et al., 2019). Formally, the DGL is defined as a mapping from a gradient $g$ to a synthetic sample $x_g^*$:

$$x_g^* = G_r(g) \triangleq \underset{x \in \mathcal{X}}{\arg\min} \left\{ L_I(x; g) \triangleq \|\nabla_\theta L(x, \theta) - g\|^2 \right\}, \tag{1}$$

where $L_I$ is the *inversion loss* matching two gradients, $\|\cdot\|$ denotes the $\ell_2$-norm either for a vector or a matrix. The $L_2$-norm of a matrix $A$ is defined by the induced norm $\|A\| = \sup_{x \neq 0} (\|Ax\| / \|x\|)$. The Euclidean distance induced by $\ell_2$-norm can be replaced by other distances, e.g., cosine similarity (Geiping et al., 2020), which is out of our scope, though.

Our goal is to understand *how* the gradient of a deep network at input $x_0$ encloses the information of $x_0$. For a linear model, e.g., $L(x, \theta) = x^\top \theta$, the gradient is $x$ and directly exposes the direction of the private sample. However, the task gets complicated when a deep neural network is used and $L(x, \theta)$ is a nonlinear loss function. The complicated functional makes exactly solving the minimization in Eq. (1) highly non-trivial and requires intensive engineering in hyperparameters and computation (Huang et al., 2021). Therefore, analyses tied to a specific DGL algorithm may not be generalizable. We introduce an assumption of a *perfect attacker* to achieve a generalizable analysis:

**Assumption 3.1.** *Given a gradient vector $g$, there is only a unique minimizer for $L_I(x; g)$.*

Since $\nabla_\theta L(x, \theta)$ is a minimizer for $G_r(\nabla_\theta L(x, \theta))$, Assumption 3.1 implies that $G_r(\nabla_\theta L(x, \theta))$ is a perfect inversion attack that recovers $x$ exactly. The assumption of a perfect attacker considers the worst-case. It allows us to develop a concrete analysis, even though such an attack may not always be feasible in practice, for example, in non-linear deep networks. To start, we outline the desired properties for our privacy analysis. **(1) Efficiency**: Privacy evaluation should be efficient in terms of computation and memory; **(2) Proximity**: The alternative should provide a good approximation or a lower bound of the risk, at least in the high-risk region; **(3) Generality**: The evaluation should be general for different models, datasets, and attacks.

## 3.1 Perturbing the Private Gradient

To figure out the association between the leakage and the gradient $g$, we formalize a counterfactual: what kind of defense can diminish the leakage? A general noise-based defense can be written as $g = \nabla_\theta L(x_0, \theta) + \delta$ where $\delta$ is a small perturbation. According to Assumption 3.1, a zero $\delta$ is not private. Instead, we are interested in non-zero $\delta$ and the recovery error (RE) of the recovered image:

$$\mathrm{RE}(x_0, G_r(g_0 + \delta)) \triangleq \|x_0 - G_r(g_0 + \delta)\| \tag{2}$$

For further analysis, we make two common and essential assumptions as follows.

**Assumption 3.2.** $L(x, \theta)$ *is twice-differentiable w.r.t.* $x$ *and* $\theta$, *i.e.,* $J \triangleq \nabla_x \nabla_\theta L(x, \theta) \in \mathbb{R}^{d_x \times d_\theta}$ *exist, where* $d_x$ *and* $d_\theta$ *are the dimensions of* $x$ *and* $\theta$, *respectively.*

**Assumption 3.3.** $JJ^\top$ *is invertible.*

We then approximate $G_r(g_0 + \delta)$ by the first-order Taylor expansion:

$$G_r(g_0 + \delta) \approx G_r(g_0) + \frac{\partial G_r(g_0)}{\partial g_0}\delta = x_0 + \frac{\partial G_r(g_0)}{\partial g_0}\delta. \tag{3}$$

By the implicit function theorem, we can show that $\frac{\partial G_r(g_0)}{\partial g_0} = (JJ^\top)^{-1}J$ (proof in Appendix B.1). Thus, for a small perturbation $\delta$, we can approximate the privacy leakage through DGL by

**Inversion Influence Function** ($\mathrm{I}^2\mathrm{F}$): $\quad \|G_r(g_0 + \delta) - x_0\| \approx \mathcal{I}(\delta; x_0) \triangleq \left\|(JJ^\top)^{-1}J\delta\right\|.$ (4)

The $\mathrm{I}^2\mathrm{F}$ includes a matrix inversion, computing which may be expensive and unstable for singular matrixes. Thus, we use a tractable lower bound of $\mathrm{I}^2\mathrm{F}$ as:

$$\left\|(JJ^\top)^{-1}J\delta\right\| \geq \frac{\|J\delta\|}{\lambda_{\max}(JJ^\top)} \triangleq \mathcal{I}_{\mathrm{lb}}(\delta; x_0), \tag{5}$$

where $\lambda_{\max}(A)$ denotes the maximal eigenvalues of a matrix $A$. Computing the maximal eigenvalue is usually much cheaper than the matrix inversion. We note that the approximation uses Assumption 3.1 and provides a general risk measurement of different attacks.

The lower bound describes how risky the gradient is in the worst case. This follows the similar intuition of Differential Privacy (DP) (Dwork, 2006) to bound the max chance (worst case) of identifying a private sample from statistic aggregation. The proposed lower bound differentiates the DP worst case in that it leverages the problem structure of gradient inversion and therefore characterizes the DGL risk more precisely.

**Efficient Evaluation.** The evaluation of $\mathcal{I}$ or its lower bound $\mathcal{I}_{\mathrm{lb}}$ is non-trivial due to three parts: computation of the Jacobian, the matrix inversion, and eigenvalue. The former two computations can be efficiently evaluated using established techniques, for example, (Koh and Liang, 2017). *1) Efficient evaluation of* $J\delta$. Instead of computing the dense Jacobian, an efficient alternative is to leverage the Jacobian-vector product, which is similar to the Hessian-vector product. We can efficiently evaluate the product of Jacobian and $\delta$ by rewriting the product as $J\delta = \nabla_x(\nabla_\theta^\top L(x_0, \theta)\delta)$. Since $\nabla_\theta^\top L(x_0, \theta)\delta$ is a scalar, the second derivative w.r.t. $x$ can be efficiently evaluated by autograd tools, e.g., PyTorch. The computation complexity is equivalent to two times of gradient evaluation in addition to one vector production. *2) Efficient matrix inversion.* We can compute $b \triangleq (JJ^\top)^{-1}J\delta$ by solving $\min_b \frac{1}{2}\left\|\delta - J^\top b\right\|^2$, whose gradient computation only contains two Jacobian-vector products. There are other alternative techniques, for example, Neumann series or stochastic approximation, as discussed in (Koh and Liang, 2017). For brevity, we discuss these alternatives in Appendix A.

In this paper, we use the *least square trick* and solve the minimization by gradient descent directly. *3) Efficient evaluation of privacy with the norm of Jacobian.* Because that $\|J\| = \sqrt{\lambda_{\max}(JJ^\top)}$, we need to compute the maximum eigenvalue of $JJ^\top$, which can be done efficiently by power iteration and the trick of Jacobian-vector products.

**Complexity.** Suppose the computation complexity for evaluating gradient is a constant in the order $\mathcal{O}(d)$ where $d$ is the maximum between the dimension of $\theta$ and $x$. Then evaluating the Jacobian-vector product is of complexity $\mathcal{O}(d)$. By the least square trick with $T$ iterations, we can compute $\mathcal{I}$ in $\mathcal{O}(dT)$ time. The lower bound $\mathcal{I}_{\text{lb}}$ replaces the computation of matrix inversion with computation on the max eigenvalue. By $T'$-step power iteration, the complexity is $\mathcal{O}(d(T + T'))$.

**Case study on Gaussian perturbation.** Gaussian noise is induced from DP-SGD (Abadi et al., 2016) where gradients are noised to avoid privacy leakage. Assuming the perturbation follows a standard Gaussian distribution, we can compute the expectation as follows:

$$\mathbb{E}[\text{RE}^2(x_0, G_r(g_0 + \delta))] \approx \mathbb{E}[\mathcal{I}^2(\delta; x_0)] = \sum_{i=1}^{d} \lambda_i^{-1}(J^\top J), \tag{6}$$

where $\lambda_i(J^\top J)$ is the $i$-th eigenvalue of $J^\top J$. The above illustrates the critical role of the Jacobian eigenvalues. For a perturbation to offer a good defense and yield a large recovery error, the Jacobian should have at least one small absolute eigenvalue. In other words, the protection is great when the Jacobian is nearly singular or rank deficient. The intuition for the insight is that a large eigenvalue indicates the high sensitivity of gradients on the sample, and therefore an inaccurate synthesized sample will be quickly gauged by the increased inversion loss.

## 3.2 Theoretic Validation

The proposed I²F assumes that the perturbation is small enough for an accurate approximation, and yet it is interesting to know how good the approximation is for larger $\delta$. To study the scenario, we provide a lower bound of the inversion error on a non-infinitesimal $\delta$, using weak Lipschitz assumptions for any two samples $x$ and $x'$.

**Assumption 3.4.** *There exists $\mu_J > 0$ such that $\|\nabla_x\nabla_\theta L(x, \theta) - \nabla_x\nabla_\theta L(x', \theta)\| \leq \mu_J \|x - x'\|$.*

**Assumption 3.5.** *There exists $\mu_L > 0$ such that $\|\nabla_\theta L(x, \theta) - \nabla_\theta L(x', \theta)\| \leq \mu_L \|x - x'\|$.*

**Theorem 3.1.** *If Assumption 3.4 and 3.5 hold, then the recovery error satisfies:*

$$\|x_0 - G_r(g_0 + \delta)\| \geq \frac{\|J\delta\|}{\mu_L \|J\| + 2\mu_J \|g_0 + \delta\|}, \tag{7}$$

*where $J = \nabla_x\nabla_\theta L(x_0, \theta)$.*

The proof is available in Appendix B.2. Similar to I²F, the inversion error could be lower bound using the Jacobian matrix. Eq. (7) matches our lower bound $\mathcal{I}_{\text{lb}}$ in Eq. (5) with an extra term of gradients. The bound implies that our approximation could be effective as a lower bound even for larger $\delta$. Besides, the lower bound provide additional information when $\delta$ is large: when $\|\delta\|$ gets infinitely large, the bound converges to $\Omega(\|J\delta\| / (\mu_J \|\delta\|))$, which does not scale up to infinity. The intuition is that the $x$ is not necessarily unbounded for matching an unbounded 'gradient'.

## 4 Empirical Validation and Extensions

We empirically show that our metric is general for different attacks with different models on different datasets from different modalities.

**Setup.** We evaluate our metric on two image-classification datasets: MNIST (LeCun, 1998) and CI-FAR10 (Krizhevsky et al., 2009). We use a linear model and a non-convex deep model ResNet18 (He et al., 2015a) (RN18) trained with cross-entropy loss. We evaluate our metric on two popular attacks: the DGL attack (Zhu et al., 2019) and the GS attack (Geiping et al., 2020). DGL attack minimizes the $L_2$ distance between the synthesized and ground-truth gradient, while the GS attack maximizes the cosine similarity between the synthesized and ground-truth gradient. We use the Adam optimizer with a learning rate of 0.1 to optimize the dummy data. Per attacking iteration, the dummy data is projected into the range of the ground-truth data $[0, 1]$ with the GS attack. When the DGL attack

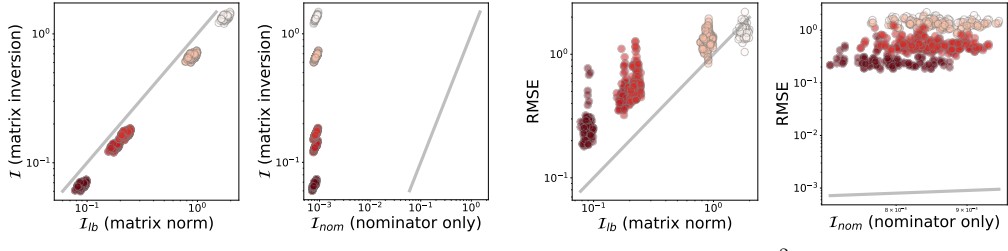

(a) Compare different approximation      (b) Compare I²F to RMSE

Figure 1: Value comparisons attacking ResNet18 on MNIST by DGL, where the grey line indicates the equal values and darker dots imply smaller Gaussian perturbation $\delta$. In (a), the y-axis is calculated as defined in Eq. (4) and $\mathcal{I}_{lb}$ is calculated as defined in Eq. (5). I²F lower bound ($\mathcal{I}_{lb}$) provides a good approximation to the exact value with matrix inversion and to the root of mean square error (RMSE) of recovered images. Instead, removing the denominator in $\mathcal{I}$ results in overestimated risks.

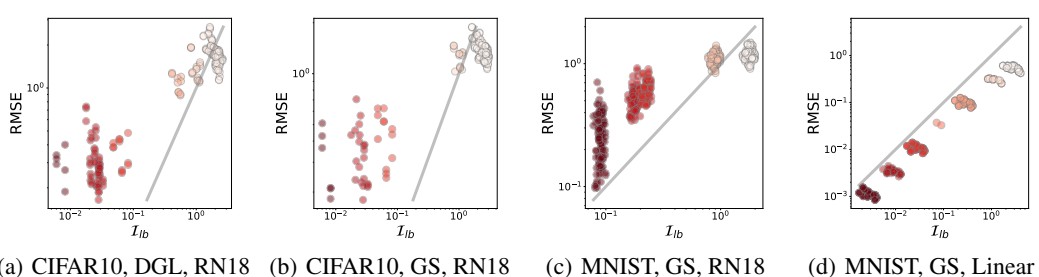

(a) CIFAR10, DGL, RN18    (b) CIFAR10, GS, RN18    (c) MNIST, GS, RN18    (d) MNIST, GS, Linear

Figure 2: I²F works under different settings: datasets, attacks, and models. The grey line indicates the equal values, and darker dots imply smaller Gaussian perturbation $\delta$.

cannot effectively recover the data, e.g., on noised linear gradients, we only conduct the GS attack to show the highest risk. We use the root of MSE (RMSE) to measure the difference between the ground-truth and recovered images. All the experiments are conducted on one NVIDIA RTX A5000 GPU with the PyTorch framework. More details and extended results are attached in Appendix C.

**Extension to singular Jacobians.** We consider the $JJ^\top$ be singular, which often happens with deep neural networks where many eigenvalues are much smaller than the maximum. In this case, the inversion of $JJ^\top$ does not exist or could result in unbounded outputs. To avoid the numerical instability, we use $\left\|(JJ^\top + \epsilon I)^{-1}J\delta\right\|$ with a constant $\epsilon$. In our experiments, the $\epsilon$ is treated as a hyperparameter and will be tuned on samples to best fit the linear relation between RMSE and $\mathcal{I}$.

**Batch data.** In practice, an attacker may only be able to carry out batch gradient inverting, $\left\|\frac{1}{n}\sum_i^n \nabla_\theta L(x_i, \theta) - \frac{1}{n}\sum_i^n g_i\right\|$, which can be upper bounded by $\frac{1}{n}\sum_i^n \left\|\nabla_\theta L(x_i, \theta) - g_i\right\|$. Such decomposition can be implemented following (Wen et al., 2022), which modifies the parameters of the final layer to reduce the averaged gradient to an update of a single sample. Thus, we only consider the more severe privacy leakage by individual samples instead of a batch of data.

**How well are the influence approximations?** In Fig. 1(a), we compare the two approximations of I²F. $\mathcal{I}_{lb}$ relaxes the matrix inversion in $\mathcal{I}$ to the reciprocal of $\left\|JJ^\top\right\|$. $\mathcal{I}_{nom}$ further drops the denominator and only keep the nominator $\|J\delta\|$. The two figures show that $\mathcal{I}_{lb}$ serves as a tight approximation to the real influence while $\mathcal{I}_{nom}$ is much smaller than the desired value. In Fig. 1(b), we see that the relaxation $\mathcal{I}_{lb}$ is also a good approximation or lower bound for the RMSE.

**Validation on different models, datasets, and attacks.** In Fig. 2, we show that I²F can work approximately well in different cases though the approximation accuracy varies by different cases. *(1) Attack:* In Figs. 2(a) and 2(b), we compare DGL to the Gradient Similarly (GS) (Geiping et al., 2020) that matches gradient using cosine similarity instead of the $L_2$ distance. We see that different attacks do not differ significantly. *(2) Dataset:* As shown in Figs. 2(b) and 2(c), different datasets could significantly differ regarding attack effectiveness. The relatively high RMSE implies that CIFAR10 tends to be harder to attack due to more complicated features than MNIST. Our metric suggests that there may exist a stronger inversion attack causing even lower RMSE. *(3) Model:*

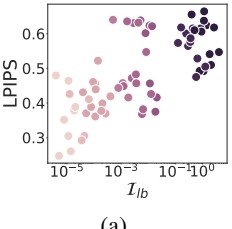

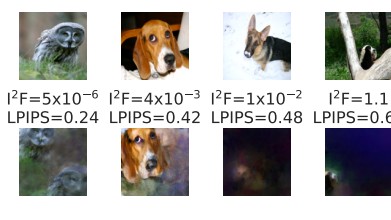

I²F=5x10⁻⁶  I²F=4x10⁻³  I²F=1x10⁻²  I²F=1.1
LPIPS=0.24  LPIPS=0.42  LPIPS=0.48  LPIPS=0.66

(a)                                    (b)

Figure 3: Evaluation of I²F on ResNet152 and ImageNet. (a): Darker color means larger noise variance. LPIPS is used to evaluate the semantic information of the recovered and original images. $I_{lb}$ is a good estimator of the semantic distance between the recovered images and original images. (b): Original (top) and recovered (bottom) images with their corresponding I²F and LPIPS. Images with a lower I²F also have a smaller LPIPS, which implies a better reconstruction.

Comparing Figs. 2(c) and 2(d), the linear model presents a better correlation than the ResNet18 (RN18). Because the linear model is more likely to be convex, the first-order Taylor expansion in Eq. (3) can approximate its RMSE than one of the deep networks.

**Results on Large Models and Datasets.** We also evaluate our I²F metric on ResNet152 with ImageNet. For larger models, the RMSE is no longer a good metric for the recovery evaluation. Even if state-of-the-art attacks are used and the recovered image is visually similar to the original image in Fig. 3(a), the two images are measured to be different by RMSE, due to the visual shift: The dog head is shifted toward the left side. To capture such shifted similarity, we use LPIPS (Zhang et al., 2018) instead, which measures the semantic distance between two images instead of the pixel-to-pixel distance like RMSE. Fig. 3(a) shows that I2F is correlated to LPIPS using large models and image scales. This implies that I2F is a good estimator of recovery similarity. In Fig. 3(b), original images with a lower I2F also have a smaller LPIPS, which means a better reconstruction. Recovered images on the right (the German shepherd and the panda) cannot be recognized while those on the left (the owl and the beagle) still have enough semantic information for humans to recognize.

**Results on Language Models and Datasets.** Since the input space of language models is not continuous like images, current attacks on images cannot be directly transferred to text. Besides, continuously optimized tokens or embeddings should be projected back to the discrete text, which induces a gap between the optimal and the original text. Thus, the investigation of privacy risks in text is an interesting yet non-trivial problem. We evaluate the proposed I²F metric on BERT (Devlin et al., 2018) and GPT-2 (Radford et al., 2019) with TAG attack (Deng et al., 2021), which minimizes $L_2$ and $L_1$ distance between gradients of the original and recovered images. We use *ROUGE-L* (Lin, 2004) and *Google BLEU* (Wu et al., 2016) to measure the semantic similarity between the original text and the recovered text. More experimental details and results with *ROUGE-1* and *feature MSE* are provided in C.2. The results are presented in Fig. 4. A darker color indicates a larger noise variance. Since the *ROUGE-L* and *Google BLEU* measure the semantic similarity of the text pair while our metric estimates the difference, two semantic metrics are negatively correlated to $\mathcal{I}_{lb}$. The results demonstrate the effectiveness of our metric in evaluating the privacy risks of discrete data.

## 5 When Does Privacy Leakage Happen?

Guided by the formulation of I²F, we provide insights on when privacy leakage happens based on the proposed metric. Our insights are rooted in the properties of Jacobian. Since we want to consider the whole property of the Jacobian matrix, we choose a shallow convolutional neural network LeNet (LeCun et al., 1998) to trade off the utility and efficiency of experiments by default.

### 5.1 Perturbations Directions Are Not Equivalent

**Implication on choosing $\delta$.** Eq. (4) clearly implies that the perturbation is not equal in different directions. Decomposing $J = U\Sigma V^\top$ using Singular Value Decomposition (SVD), we obtain $\mathcal{I}(\delta; x_0) = \left\| U\Sigma^{-1}V^\top \delta \right\|$. Thus, $\delta$ tends to yield a larger I²F value if it aligns with the directions of small eigenvalues of $JJ^\top$.

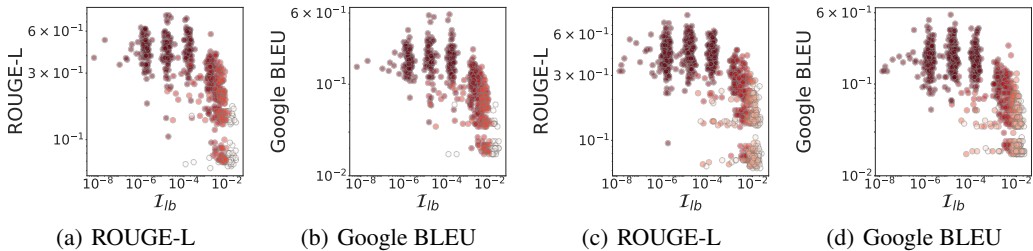

| (a) ROUGE-L | (b) Google BLEU | (c) ROUGE-L | (d) Google BLEU |

Figure 4: Evaluation of $\mathcal{I}_{lb}$ on BERT (a-b) and GPT-2 (c-d). A darker color means a larger noise variance. ROUGE-L and Google BLEU are used to evaluate the semantic similarity between the original text and the recovered text. $\mathcal{I}_{lb}$ is linearly correlated to the two semantic metrics, which means $\mathcal{I}_{lb}$ can be used to estimate the privacy risk of the private text.

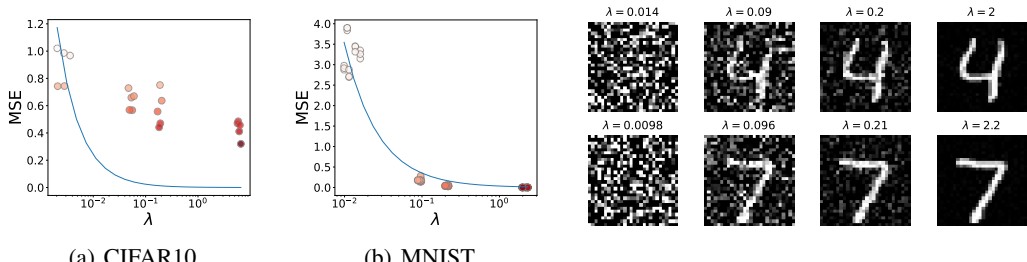

| (a) CIFAR10 | (b) MNIST |

Figure 6: Same perturbation sizes but different protection effects by different eigenvectors. In (a) and (b), MSEs of DGL attacks are reversely proportional to eigenvalues on the LeNet model. Blue curves are scaled $1/\lambda$. Darker dots indicate smaller MSE (higher risks). Recovered MNIST images associated with different eigenvalues are present on the right.

**Singular values of Jacobian of deep networks.** In Fig. 5, we examine the eigenvalues of LeNet on the MNIST and CIFAR10 datasets. On both datasets, there are always a few eigenvalues of $JJ^\top$ (around $10^{-2}$) that are much smaller than the largest one ($\geq 1$). Observing such a large gap between $JJ^\top$ eigenvalues, it is natural to ask: how do the maximal/minimal eigenvalues affect the DGL attacks?

Figure 5: The distribution of eigenvalues of $JJ^\top$ of two datasets on the LeNet model.

**Comparing eigenvectors in defending DGL.** We consider a special case of perturbation by letting $\delta$ be an eigenvector of $JJ^\top$. Then the I²F will be $1/\lambda$ where $\lambda$ is the corresponding eigenvalue. We conjecture $1/\lambda$ could predict the MSE of DGL attacks. To verify the conjecture, we choose 4 eigenvectors with distinct eigenvalues per sample. The results for the LeNet model are present in Fig. 6. We see that the MSE decreases by $\lambda$. For the MNIST dataset, the MSE-$\lambda$ relation is very close to the predicted $1/\lambda$. Though the curve is biased from the ground truth for CIFAR10, we still can use $1/\lambda$ to lower bound the recovery error. The bias in CIFAR10 is probably due to the hardness of recovering the more complicated patterns than the digit images. The recovered images in Fig. 6 suggest that even with the same perturbation scale, there exist many bad directions for defense. In the worst case, the image can be fully covered. The observation is an alerting message to the community: *protection using random noise may leak private information*.

## 5.2 Privacy Protection Could Be Unfair

Our analysis of the Gaussian perturbation in Eq. (6) indicates that the privacy risk hinges on the Jacobian and therefore the sample. We conjecture that the resulting privacy risk will vary significantly due to the sample dependence. In Fig. 7, we conduct fine-grained experiments to study the variance of privacy protection, when Gaussian perturbation on gradients is used for protection. We use a well-trained model which is thought to be less risky than initial models, for example, in (Balunović et al., 2022). Though the average of MSE implies a reasonable privacy degree as reported in previous literature, the large variance delivers the opposite message that some samples or classes are not that

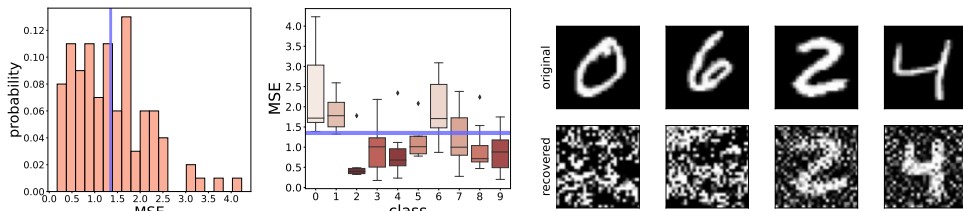

Figure 7: The sample-wise and class-wise statistics of the DGL MSE on the MNIST dataset, when gradients are perturbed with Gaussian noise of variance $10^{-3}$. The purple lines indicate the average values. Large variances are observed among samples and classes. The original (top) and recovered (bottom) images for the well- and poorly-protected classes are depicted on the right side.

safe. In the sense of samples, many samples are more vulnerable than the average. For the classes, some classes are obviously more secure than others. Thus, when the traditional metric focusing on average is used, it may deliver a fake sense of protection unfairly for specific classes or samples.

To understand which kind of samples could be more vulnerable to attack, we look into the Jacobian and ask when $JJ^\top$ will have larger eigenvalues and higher risks in Fig. 7. Consider a poisoning attack, where an attacker aims to maximize the training loss by adding noise $\delta_x$ to the sample $x$. Suppose the parameter is updated via $\theta'(x+\delta_x) = \theta - \nabla_\theta L(x+\delta_x, \theta)$. Let $J_\delta = \nabla_x \nabla_\theta L(x_1 + \delta_x, \theta)$ and we can derive the maximal loss amplification on a test sample $x_1$ when perturbing $x$ as:

$$\max_{\|\delta_x\| \leq 1} \mathbb{E}_{x_1}[L(x_1, \theta'(x+\delta_x))] \approx \max_{\|\delta_x\| \leq 1} \mathbb{E}_{x_1}[L(x_1, \theta'(x))] + \mathbb{E}_{x_1}[\delta_x^\top \nabla_x L(x_1, \theta'(x))]$$

$$= \max_{\|\delta_x\| \leq 1} \delta_x^\top J_\delta \mathbb{E}_{x_1}[\nabla_{\theta'} L(x_1, \theta')]$$

$$= \|J_\delta \mathbb{E}_{x_1}[\nabla_{\theta'} L(x_1, \theta')]\|$$

$$\leq \|J_\delta\| \|\mathbb{E}_{x_1}[\nabla_{\theta'} L(x_1, \theta')]\|.$$

As shown by the inequality, the sample with large $\|J_\delta\|$ may significantly bias the training after mild perturbation $\delta_x$. If Assumption 3.4 holds, then $\|J\|$ can be associated with $\|J_\delta\|$ by $|\|J\| - \|J_\delta\|| \leq \mu_J$. Now we can connect the privacy vulnerability to the data poisoning. With Eq. (6), samples that suffer high privacy risks due to large $\|J\|$ could also be influential in training and can be easily used to poison training.

### 5.3 Model Initialization Matters

Since Jocabian leans on the parameters, we hypothesize that the way we initialize a model can also impact privacy. Like previous work (Sun et al., 2020; Balunović et al., 2022; Zhu et al., 2019), we are interested in the privacy risks when a model is randomly initialized without training. Unlike previous work, we further ask which initialization strategy could favor privacy protection under the same Gaussian perturbation on gradients.

**Case Study: One-layer network.** A one-layer network with nonlinear activation $\sigma(\cdot)$ is a simple yet sufficient model for analyzing initialization. Let $L(x, \theta) = \frac{1}{2} \left\| \sigma(\theta^\top x) - b \right\|^2$, where $b$ is a constant. Denote the product $\theta^\top x$ as $a$. The Jacobian is given by:

$$J = \frac{\partial^2 L}{\partial x \partial \theta} = \frac{\partial \sigma}{\partial a} \left\{ (\sigma(\theta^\top x) - b)I + x\theta^\top \right\}.$$

(a) MNIST    (b) CIFAR10

Figure 8: Different initialization strategies could result in distinct MSEs.

(1) Apparently, a well-trained $\theta$ with $(\sigma(\theta^\top x) - b) = 0$ will cause $J$ to be a rank-one matrix. Note that the gradient will be zero in such a case and no longer provide any information about the private sample. Consistent with our theoretical results, where the inverse Jacobian makes $\mathcal{I}$ infinite, the gradient will be perfectly private at any perturbation. (2) If $\sigma(\theta^\top x) - b$ is non-zero, the singular values of $J$ will depends on how close $\mathbb{E}[\sigma(\theta^\top x)]$ is to $b$. If $\mathbb{E}[\sigma(\theta^\top x)]$ approaches $b$ at initialization, then $J$ will be approximately singular, and therefore the initialization enhances privacy preservation.

**Comparing initialization.** To evaluate the effect of initialization on the inversion, we conduct experiments of the LeNet model on MNIST and CIFAR10 datasets with DGL and GS attacks. We add Gaussian noise to the gradients, which implies an expected MSE should be proportional to $\mathbb{E}[\mathcal{I}^2] = \sum_i \lambda_i^{-1}$ as proved in Eq. (6). Here, we evaluate how the initialization will change eigenvalues of $JJ^\top$ and thus change the MSE of DGL. Four commonly used initialization techniques are considered here: uniform distribution in range $[-0.5, 0.5]$, normal distribution with zero mean and variance $0.5$, kaiming (He et al., 2015b) and xavier (Glorot and Bengio, 2010) initialization. For each initialization, the same 10 samples are used to conduct the attack three times. The results are shown in Fig. 8. We observe a significant gap between initialization mechanisms. Using uniform initialization cast serious risks of leaking privacy under the same Gaussian defense. Though not as significant as uniform initialization, the normal initialization is riskier than rest two techniques. kaiming and xavier methods can favor convergence in deep learning and here we show that they are also preferred for privacy. A potential reason is that the two methods can better normalize the activations to promote the Jacobian singularity.

In parallel with our work, Wang et al. (2023a) also finds that the initialization is impactful on inversion attacks. Despite their analysis being based on a different metric, the layer-wise variance of the model weights, their work and ours have a consistent conclusion that models initialized with a uniform distribution face higher privacy risks than that with a kaiming distribution

## 6   Conclusion and Discussion

In this paper, we introduce a novel way to use the influence functions for analyzing Deep Gradient Leakage (DGL). We propose a new and efficient approximation of DGL called the Inversion Influence Function ($I^2F$). By utilizing this tool, we gain valuable insights into the occurrence and mechanisms of DGL, which can greatly help the future development of effective defense methods.

**Limitations.** Our work may be limited by some assumptions and approximations. First, we worked on the worst-case scenario where a strong attack conducts perfect inversion attacks. In practice, such an assumption can be strong, especially for highly complicated deep networks. The gap between existing attacks and perfect attacks sometimes leads to a notable bias. However, we note that recent years witnessed many techniques that significantly improved attacking capability  (Geiping et al., 2020; Jeon et al., 2021; Zhao et al., 2020), and our work is valuable to bound the risks when the attacks get even stronger over time. Second, similar to the traditional influence function, $I^2F$ can be less accurate and suffers from large variance in extremely non-convex loss functions. Advanced linearization techniques (Bae et al., 2022) can be helpful in improving the accuracy of influence. Then extending our analysis to bigger foundation models may bring intriguing insights into the scaling law of privacy.

**Future Directions.** As the first attempt at influence function in DGL, our method can serve multiple purposes to benefit future research. For example, our metric can be used to efficiently examine the privacy breach before sending gradients to third parties. Since $I^2F$ provides an efficient evaluation of the MSE, it may be directly optimized in conjunction with the loss of main tasks. Such joint optimization could bring in the explicit trade-off between utility and privacy in time. In comparison, traditional arts like differential privacy are complicated by tuning the privacy parameter for the trade-off. Furthermore, we envision that many techniques can be adopted to further enhance the analysis. For example, unrolling-based analysis leverages the iterative derivatives in the DGL to uncover the effectiveness of gradient perturbations (Pruthi et al., 2020).

**Broader Impacts.** Data privacy has been a long-term challenge in machine learning. Our work provides a fundamental tool to diagnose privacy breaches in the gradients of deep networks. Understanding when and how privacy leakage happens can essentially help the development of defenses. For example, it can be used for designing stronger attacks, which leads to improved defense mechanisms and ultimately benefit the privacy and security of machine learning.

## 7   Acknowledgments

This research was supported by the National Science Foundation (IIS-2212174, IIS-1749940), National Institute of Aging (IRF1AG072449), and the Office of Naval Research (N00014- 20-1-2382).

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

# A Method

## A.1 Other Efficient Evaluation Techniques

The techniques for efficiently evaluating implicit gradients can be referred to (Koh and Liang, 2017; Grazzi et al., 2020). As computing the inverse second-order derivatives is the most computation-intensive operation, we will focus on it. Here, we briefly summarize two supplementary techniques introduced in Section 3.1.

**Conjucate gradient.** In Section 3.1, we use the trick of least square to compute the $(JJ^\top)^{-1}J\delta$. When $JJ^\top \succ 0$, we can solve the least square problem by the conjugate gradient (CG) method, which only needs $\mathcal{O}(d)$ time to converge. However, the algorithm will be unstable when the matrix $JJ^\top$ is ill-conditioned. As observed in Fig. 5, the $JJ^\top$ is likely to be ill-conditioned for deep networks. But here we provide this alternative for linear models such that $J$ can be evaluated faster.

**Neumann series.** We can leverage the Neumann series to compute the matrix inverse. By the Neumann series, we have $(JJ^\top)^{-1}J\delta = \lim_{t\to\infty}\sum_{i=0}^{t}(I - JJ^\top)^i J\delta$. Let $s_t \triangleq \sum_{i=0}^{t}(I - JJ^\top)^i J\delta$ and $s_0 \triangleq J\delta$. Then the computation can be done by iteration $s_{t+1} = (I - JJ^\top)s_t + J\delta$ which only includes Jacobian-vector products.

# B Proofs

## B.1 Proof of the Approximation by Implicit Gradients

Here, we provide the proof for $\frac{\partial G_r(g_0)}{\partial g_0} = (JJ^\top)^{-1}J$. Recall Eq. (1) as

$$x^* = G_r(g) = \arg\min_x L_I(x;g) \triangleq \|\nabla_\theta L(x,\theta) - g\|^2. \tag{8}$$

The stationary condition of the minimization gives

$$\frac{\partial L_I(x^*;g)}{\partial x^*} = 0.$$

Given a small perturbation $\Delta_g \to 0$ on the gradient, we can estimate corresponding perturbation $\Delta_{x^*} \to 0$ as a function of $\Delta_g$. Thus, we can approximate $\frac{\partial x^*}{\partial g}$ by $\frac{\partial \Delta_{x^*}}{\partial \Delta_g}$. Use Taylor expansion to show approximately

$$\frac{\partial L_I(x^*;g)}{\partial x^*} + \frac{\partial^2 L_I(x^*;g)}{\partial g \partial x^*}\Delta_g + \frac{\partial^2 L_I(x^*;g)}{\partial x^{*2}}\Delta_{x^*} \approx 0$$

$$\Delta_{x^*} \approx -\left(\frac{\partial^2 L_I(x^*;g)}{\partial x^{*2}}\right)^{-1}\frac{\partial L_I(x^*;g)}{\partial x^*}\Delta_g$$

$$\frac{\partial x_g^*}{\partial g} \approx -\left(\frac{\partial^2 L_I(x^*;g)}{\partial x^{*2}}\right)^{-1}\frac{\partial^2 L_I(x^*;g)}{\partial g \partial x^*} \tag{9}$$

where we drop higher-order perturbations. The above derivations can be rigorously proved using the Implicit Function Theorem. Since $\frac{\partial L_I(x^*;g)}{\partial x^*} = 2(\nabla_\theta L(x^*,\theta) - g)\nabla_x \nabla_\theta L(x^*,\theta)$, we can derive

$$\frac{\partial^2 L_I(x^*;g)}{\partial g \partial x^*} = -2\nabla_x \nabla_\theta L(x^*,\theta)$$

and

$$\frac{\partial^2 L_I(x^*;g)}{\partial x^{*2}} = 2(\nabla_\theta L(x^*,\theta) - g)\nabla_x^2 \nabla_\theta L(x^*,\theta) + 2\nabla_x \nabla_\theta L(x^*,\theta)(\nabla_x \nabla_\theta L(x^*,\theta))^\top.$$

As $x_0 = x^* = G_r(g_0)$ and $g_0 = \nabla_\theta L(x^*,\theta)$, we can substitute them to obtain

$$\frac{\partial^2 L_I(x_0;g)}{\partial g \partial x_0} = -2\nabla_x \nabla_\theta L(x_0,\theta) := -2J(x^*(g_0),\theta), \tag{10}$$

$$\frac{\partial^2 L_I(x_0;g)}{\partial x_0^2} = 2(g_0 - g)\nabla_x^2 \nabla_\theta L(x_0,\theta) + 2JJ^\top. \tag{11}$$

Let $g = g_0$. Combine Eqs. (9) to (11) to get

$$\frac{\partial G_r(g_0)}{\partial g_0} = (JJ^\top)^{-1}J.$$

## B.2 Proof of Theorem 3.1

Before we prove our main theorem, we prove several essential lemmas as below.

**Lemma B.1.** $\left\| \nabla_x \left\| \nabla_\theta L(x,\theta) \right\|^2 - \nabla_x \left\| \nabla_\theta L(x',\theta) \right\|^2 \right\| \leq (\mu_L \left\| \nabla_x \nabla_\theta L(x,\theta) \right\| + \mu_J \left\| \nabla_\theta L(x',\theta) \right\|) \left\| x - x' \right\|$

*Proof.*

$$\left\| \nabla_x \left\| \nabla_\theta L(x,\theta) \right\|^2 - \nabla_x \left\| \nabla_\theta L(x',\theta) \right\|^2 \right\|$$
$$= \left\| \nabla_x \nabla_\theta L(x,\theta) \nabla_\theta L(x,\theta) - \nabla_x \nabla_\theta L(x,\theta) \nabla_\theta L(x',\theta) \right.$$
$$\left. + \nabla_x \nabla_\theta L(x,\theta) \nabla_\theta L(x',\theta) - \nabla_x \nabla_\theta L(x',\theta) \nabla_\theta L(x',\theta) \right\|$$
$$\leq \left\| \nabla_x \nabla_\theta L(x,\theta) \right\| \left\| \nabla_\theta L(x,\theta) - \nabla_\theta L(x',\theta) \right\| + \left\| \nabla_x \nabla_\theta L(x',\theta) - \nabla_x \nabla_\theta L(x,\theta) \right\| \left\| \nabla_\theta L(x',\theta) \right\|.$$

Using Assumption 3.4 and 3.5 directly lead to

$$\left\| \nabla_x \left\| \nabla_\theta L(x,\theta) \right\|^2 - \nabla_x \left\| \nabla_\theta L(x',\theta) \right\|^2 \right\| \leq (\mu_L \left\| \nabla_x \nabla_\theta L(x,\theta) \right\| + \mu_J \left\| \nabla_\theta L(x',\theta) \right\|) \left\| x - x' \right\|.$$

$\square$

**Lemma B.2.** $\left\| \nabla_x(\nabla_\theta^\top L(x,\theta)g) - \nabla_x(\nabla_\theta^\top L(x',\theta)g) \right\| \leq \mu_J \left\| g \right\| \left\| x - x' \right\|$

*Proof.* By Assumption 3.4, we have

$$\left\| \nabla_x(\nabla_\theta^\top L(x,\theta)g) - \nabla_x(\nabla_\theta^\top L(x',\theta)g) \right\|$$
$$\leq \left\| \nabla_x \nabla_\theta L(x,\theta) - \nabla_x \nabla_\theta L(x',\theta) \right\| \left\| g \right\|$$
$$\leq \mu_J \left\| g \right\| \left\| x - x' \right\|.$$

$\square$

**Lemma B.3.** *The inversion loss* $L_I(x;g)$ *defined satisfies* $\left\| \nabla_x L_I(x;g) - \nabla_x L_I(x';g) \right\| \leq \mu \left\| x - x' \right\|$ *where*

$$\mu(x,x',\theta,g) = \mu_L \left\| \nabla_x \nabla_\theta L(x,\theta) \right\| + \mu_J \left\| \nabla_\theta L(x',\theta) \right\| + \mu_J \left\| g \right\|. \tag{12}$$

*Proof.* Since

$$\nabla_x L_I(x;g) = 2\nabla_x \nabla_\theta L(x,\theta)(\nabla_\theta L(x,\theta) - g)$$
$$= 2\nabla_x \left\| \nabla_\theta L(x,\theta) \right\|^2 - 2\nabla_x \nabla_\theta L(x,\theta)g,$$

we can derive

$$\left\| \nabla_x L_I(x;g) - \nabla_x L_I(x';g) \right\|$$
$$= 2 \left\| \nabla_x \left\| \nabla_\theta L(x,\theta) \right\|^2 - \nabla_x \left\| \nabla_\theta L(x',\theta) \right\|^2 - \left[ \nabla_x \nabla_\theta^\top L(x,\theta) - \nabla_x \nabla_\theta^\top L(x',\theta) \right] g \right\|$$
$$\leq 2 \left\| \nabla_x \left\| \nabla_\theta L(x,\theta) \right\|^2 - \nabla_x \left\| \nabla_\theta L(x',\theta) \right\|^2 \right\| + 2 \left\| \left[ \nabla_x \nabla_\theta^\top L(x,\theta) - \nabla_x \nabla_\theta^\top L(x',\theta) \right] g \right\|.$$

By Lemma B.1 and Lemma B.2, we have

$$\left\| \nabla_x \left\| \nabla_\theta L(x,\theta) \right\|^2 - \nabla_x \left\| \nabla_\theta L(x',\theta) \right\|^2 \right\| \leq \mu_1 \left\| x - x' \right\|,$$
$$\left\| \nabla_x(\nabla_\theta^\top L(x,\theta)g) - \nabla_x(\nabla_\theta^\top L(x',\theta)g) \right\| \leq \mu_2 \left\| x - x' \right\|.$$

where $\mu_1 = \mu_L \left\| \nabla_x \nabla_\theta L(x,\theta) \right\| + \mu_J \left\| \nabla_\theta L(x',\theta) \right\|$ and $\mu_2 = \mu_J \left\| g \right\|$. Let $\mu = 2\mu_1 + 2\mu_2$. Then we can get

$$\left\| \nabla_x L_I(x;g) - \nabla_x L_I(x';g) \right\| \leq \mu \left\| x - x' \right\|.$$

$\square$

**Theorem B.1** (Restated from Theorem 3.1). *Let $x_0$ be the private data and $g_0 \triangleq \nabla_\theta L(x_0, \theta)$ be its corresponding gradient which is treated as a constant. If Assumption 3.4 and 3.5 hold, then the square root of the recovery RMSE satisfies:*

$$\|x_0 - G_r(g_0 + \delta)\| \geq \frac{\|J\delta\|}{\mu_L \|J\| + 2\mu_J \|g_0 + \delta\|}, \tag{13}$$

*where $J = \nabla_x \nabla_\theta L(x_0, \theta)$.*

*Proof.* Utilize the stationary condition $\nabla_x L_I(x_g^*; g) = 0$ and Lemma B.3 to obtain

$$\|\nabla_x L_I(x; g)\| \leq \mu(x, x_g^*, \theta, g) \|x - x_g^*\|, \ \forall x.$$

As $x_0$ is the private sample whose gradient is $g_0 \triangleq \nabla_\theta L(x_0, \theta)$, then we have

$$\|x_0 - G_r(g_0 + \delta)\| \geq \frac{1}{\mu(x_0, x_{g_0+\delta}^*, \theta, g_0 + \delta)} \|\nabla_x L_I(x_0; g_0 + \delta)\|$$

Because

$$\nabla_x L_I(x_0; g_0 + \delta) = 2\nabla_x \nabla_\theta L(x_0, \theta)(\nabla_\theta L(x_0, \theta) - g_0 - \delta)$$
$$= 2\nabla_x \nabla_\theta L(x_0, \theta)\delta,$$

we can attain

$$\|x_0 - G_r(g_0 + \delta)\| \geq \frac{2}{\mu} \|\nabla_x \nabla_\theta L(x_0, \theta)\delta\|.$$

With $\nabla_\theta L(x_{g_0+\delta}^*, \theta) = g_0 + \delta$, we can obtain

$$\mu(x_0, x_{g_0+\delta}^*, \theta, g_0 + \delta) = 2\mu_L \|\nabla_x \nabla_\theta L(x_0, \theta)\| + 2\mu_J \|\nabla_\theta L(x_{g_0+\delta}^*, \theta)\| + 2\mu_J \|g_0 + \delta\|$$
$$= 2\mu_L \|J\| + 4\mu_J \|g_0 + \delta\|.$$

$\square$

## C  Experiments

### C.1  Experimental Details

**Model architectures.** The linear model we use is a matrix that maps the input data into a vector. The LeNet model is a convolutional neural network with 4 convolutional layers and 1 fully connected layer. We use the modified version following previous privacy papers Sun et al. (2020), whose detailed structure is in Table 1. ResNet18 is a popular deep convolutional network in computer vision with batch-normalization and residual layers (He et al., 2015a). Cross entropy loss is used as the loss function in all the experiments.

**Experimental settings.** We conduct two kinds of attacks in our paper: DGL and GS attacks. The learning rate of the two attacks is 0.1 and we use Adam as the optimizer. To consider a more powerful attack, only a single image is reconstructed in each inversion. When inverting LeNet, we uniformly initialize the model parameters in the range of $[-0.5, 0.5]$ as (Sun et al., 2020) to get a stronger attack. When inverting ResNet18, we use the default initialization method in PyTorch and follow Huang et al. (2021) to use BN statistics as an additional regularization term to conduct a stronger attack.

Table 1: A modified version of LeNet. Conv represents a convolutional layer. FC means a fully-connected layer.

| Layers |
| --- |
| Conv(in_channels=3, out_channels=12, kernel_size=5) |
| Conv(in_channels=12, out_channels=12, kernel_size=5) |
| Conv(in_channels=12, out_channels=12, kernel_size=5) |
| Conv(in_channels=12, out_channels=12, kernel_size=5) |
| Flattern |
| FC(out_features=10) |

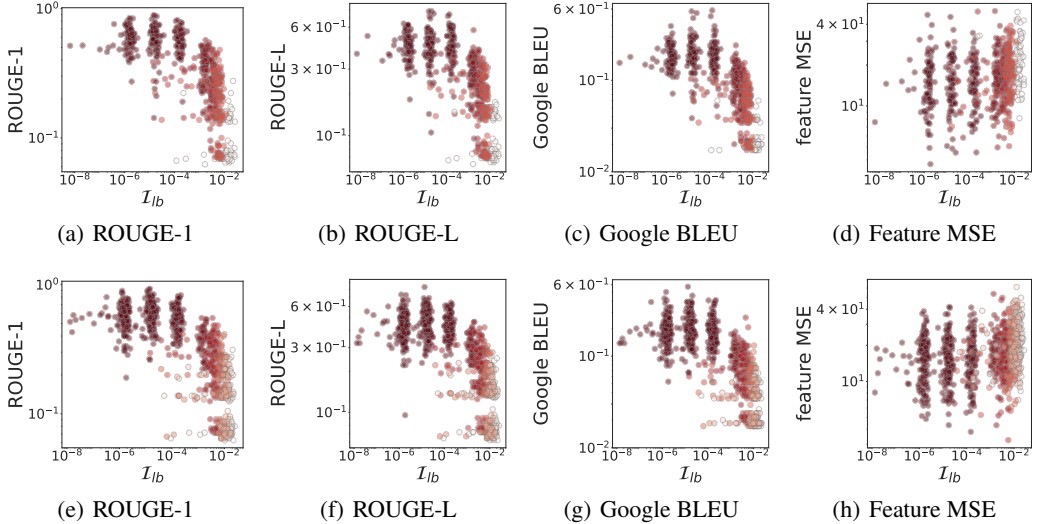

(a) ROUGE-1      (b) ROUGE-L      (c) Google BLEU      (d) Feature MSE

(e) ROUGE-1      (f) ROUGE-L      (g) Google BLEU      (h) Feature MSE

Figure 9: Evaluation of $\mathcal{I}_{lb}$ on BERT (top) and GPT-2 (bottom). A darker color means a larger noise variance. Four metrics are used to evaluate the semantic similarity between the original text and the recovered text. $\mathcal{I}_{lb}$ is linearly correlated to the four semantic metrics, which means $\mathcal{I}_{lb}$ can be used to estimate the privacy risk of the private text.

## C.2   Empirical Validation on Language Data

We evaluate the proposed I$^2$F metric on BERT (Devlin et al., 2018) and GPT-2 (Radford et al., 2019), which are popular language models in natural language processing. We use TAG (Deng et al., 2021), which is an attack on Transformer-based language models based on the $L_1$ and $L_2$ distance between the original gradient and the dummy gradient, and follow the code in `https://github.com/JonasGeiping/breaching`. We use the default setting in the code and iteratively update the input embedding 12,000 times. We randomly sample 70 sentences from WikiText-103 (Merity et al., 2016) as the private text. We use *ROUGE-1*, *ROUGE-L* (Lin, 2004), *Google BLEU* (Wu et al., 2016) and *feature MSE* to measure the semantic similarity between the original text and the recovered text. ROUGE-1 measures the overlap of 1-grams in the original and recovered text, while ROUGE-L measures the length of the longest common subsequence between two sentences. While ROUGE metrics calculate the reconstruction recall, the Google BLEU score uses as the output the smaller value of the precision and recall of the original and recovered text and has a broader range of the overlap of $n$-grams, where $n = 1, 2, 3, 4$. Since the above three metrics are discrete and not consistent with our assumptions, we include a continuous metric, the feature MSE, which measures the distance between the final layer's feature of the original and recovered text.

The results are presented in Fig. 9. A darker color indicates a larger noise variance. For each noise variance, we randomly sample the perturbation from a zero-mean Gaussian distribution 5 times and repeat this for 3 different random seeds. It shows that I$^2$F is correlated to these four metrics, which means I$^2$F can be used to estimate the privacy risk of text datasets with large language models. The correlation of BERT and GPT-2 between the four metrics has a similar mode. Though ROUGE-1, ROUGE-L and Google BLEU measure the structural similarity of sentences, which consists of discrete tokens, I$^2$F presents a clear correlation. For the feature MSE, although I$^2$F has a less distinguished correlation with feature MSE, it can still be utilized to estimate the privacy risk.

## C.3   Efficiency

To show the efficiency of computing the I$^2$F values, we compare it with the GS attack on randomly-picked CIFAR10 images with ResNet18. As the major complexity comes from evaluating the maximal eigenvalue via the power iteration method, we compare the time required for the power iteration against the inversion loss of GS to converge. We notice that the convergence depends on the initialization of the power iteration and the learning rate for DGL. Thus, we repeat power iteration with

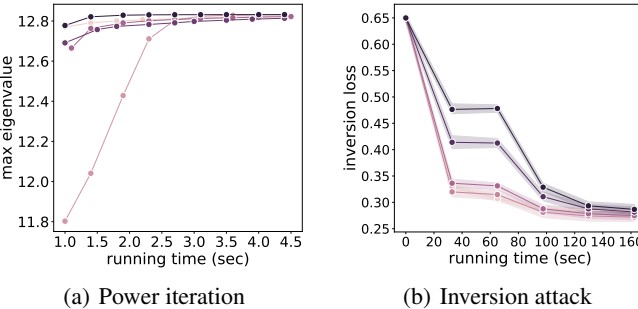

(a) Power iteration            (b) Inversion attack

Figure 10: Evaluation of the efficiency of computing $\lambda_{\max}(JJ^\top)$ (our method) by power iteration and inversion attack by minimizing inversion loss ($L_I$). Colors in (a) indicate different seeds. Darker colors in (b) indicate larger learning rates. Our method using power iteration can converge faster than direct inversion attacks.

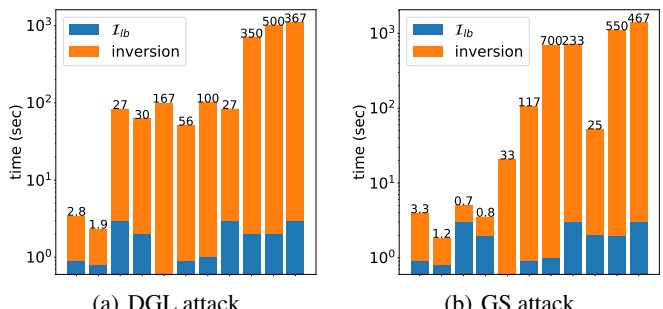

(a) DGL attack            (b) GS attack

Figure 11: Comparison of the efficiency of computing $\mathcal{I}_{lb}$ (our method) by power iteration and **inversion** attack by minimizing inversion loss ($L_I$). Blue bars indicate the time of computing $\mathcal{I}_{lb}$ while orange bars indicate the time of minimizing inversion loss by DGL and GS. The time ratio of computing $\mathcal{I}_{lb}$ versus minimizing inversion loss is present above the orange bars. The x-axis are model-dataset pairs sorted by the model scales: MLP-MNIST, MLP-CIFAR10, LeNet-MNIST, LeNet-CIFAR10, RN18-MNIST, RN18-CIFAR10, RN34-CIFAR10, RN50-CIFAR10, RN101-CIFAR10, RN152-CIFAR10, RN152-ImageNet. For large models and datasets, where minimizing inversion loss needs a huge computation overhead, $\mathcal{I}_{lb}$ can provide an efficient estimation of the privacy risk.

5 different seeds. For the inversion attack, we evaluate multiple learning rates $(1, 0.5, 0.1, 0.05, 0.01)$ to show the fastest convergence. Each experiment is repeated 5 times with different random seeds. As shown in Fig. 10, the power iteration method can converge to the maximal eigenvalue within 50 iterations (5 seconds at most). In comparison, the inversion loss demands 3000 more iterations in 150 seconds to fully converge, which is 20 times larger than the power iteration method. Thus, our method can give an accurate and fast approximation of the recovery MSE without the exhaustive whole inversion process.

In Fig. 11, we compare the computation cost of computing $\mathcal{I}_{lb}$ with minimizing inversion loss. We show that for almost all the models and datasets we evaluate, the time ratio is larger than $1$, which means it is more efficient to compute $\mathcal{I}_{lb}$ than minimize the inversion loss. It indicates that our method is a more efficient way to estimate the privacy risk for most models and datasets, than the empirical inversion attack. Another key point is that, for large models and datasets, such as models larger than RN18 with CIFAR10 or ImageNet, the time ratio is even much larger than $500$. When the time consumption of inversion attacks on these models and datasets is huge (about 16 minutes or even longer), our method significantly reduces the computation overheads for estimating the privacy risks.

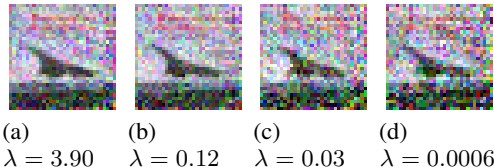

(a)       (b)       (c)       (d)
$\lambda = 3.90$   $\lambda = 0.12$   $\lambda = 0.03$   $\lambda = 0.0006$

Figure 12: Same perturbation sizes but different protection effects by different eigenvectors of LeNet. Recovered CIFAR10 images associated with different eigenvectors are present. When perturbing with eigenvectors with smaller eigenvalues, the recovered images are more noisy.

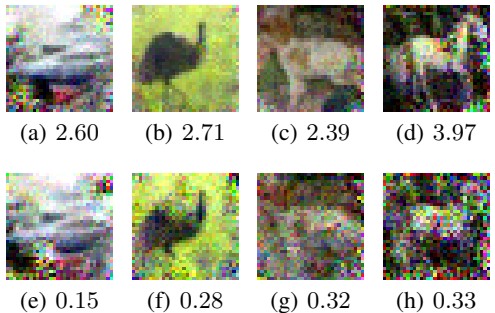

(a) 2.60    (b) 2.71    (c) 2.39    (d) 3.97

(e) 0.15    (f) 0.28    (g) 0.32    (h) 0.33

Figure 13: Same perturbation sizes but different protection effects by different eigenvectors of ResNet18. Recovered CIFAR10 images associated with different eigenvalues are present. When perturbing with eigenvectors with smaller eigenvalues, the recovered images are more noisy and lack some semantic information.

### C.4   More Visual Results

**More images of unequal perturbations.** We present in Fig. 12 the recovered CIFAR10 images when perturbing the gradient with eigenvectors with different eigenvalues. When perturbing with eigenvectors with smaller eigenvalues, the recovered images are more noisy, which is consistent with our former observation.

We also present the unequal perturbations with different eigenvectors of ResNet18 on the CIFAR10 dataset in Fig. 13. Even with the same perturbation scale, the eigenvectors with larger eigenvalues provide stronger protection, where the corresponding recovered images are more noisy and lose some semantic information.

**More images of unfair privacy protection.** We show more results of unfair privacy protection in Fig. 14. The images of digits 5 and 8 can still be recognized by their outlines, while images of digits 7 and 9 are unrecognizable noise.

We also show the unfair privacy protection of ResNet18 on the CIFAR10 dataset in Fig. 15. In this experiment, we also observe a large variance of recovery MSE among samples, indicating sample-wise unfairness. At the class level, we still can find gradients of a few classes to be easily inverted. For example, class 8 has most MSEs lower than the average value.

## D   $\text{I}^2\text{F}$ with Gradient Pruning Defense

We present the relationship between the RMSE and $\mathcal{I}_{lb}$ in Fig. 16. The y-axis is RMSE and the x-axis is $\mathcal{I}_{lb}$. It shows $\mathcal{I}_{lb}$ can be used to estimate the worst-case privacy risk with gradient pruning defense.

## E   Comparison of $\text{I}^2\text{F}$ with More Metrics

MSE is a pixel-wise distance that lacks semantic and structural information. To evaluate the effectiveness of $\text{I}^2\text{F}$ on more metrics, we consider SSIM and LPIPS (Zhang et al., 2018) to measure the

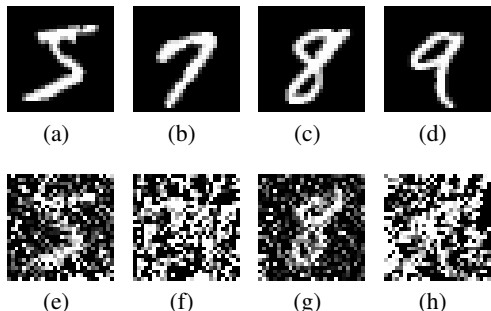

Figure 14: Original (top) and corresponding recovered (bottom) images of LeNet on the MNIST dataset. The gradients are perturbed with Gaussian noise of variance $10^{-3}$. The defense is unfair as images of digit 5 and digit 8 can be recognized by the outline.

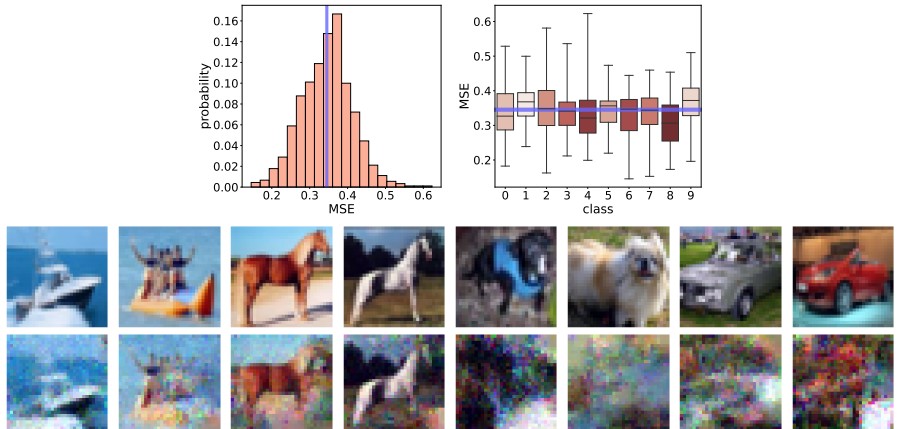

Figure 15: The sample-wise and class-wise statistics of the GS MSE on the CIFAR10 dataset of ResNet18. The purple lines indicate the average values. Large variances are observed among samples. The original (first row) and recovered (second row) images for the well- and poorly-protected classes are depicted at the bottom. The defense is unfair as some classes, e.g., class 7 (horse) and class 8 (ship), are more vulnerable to inversion attacks.

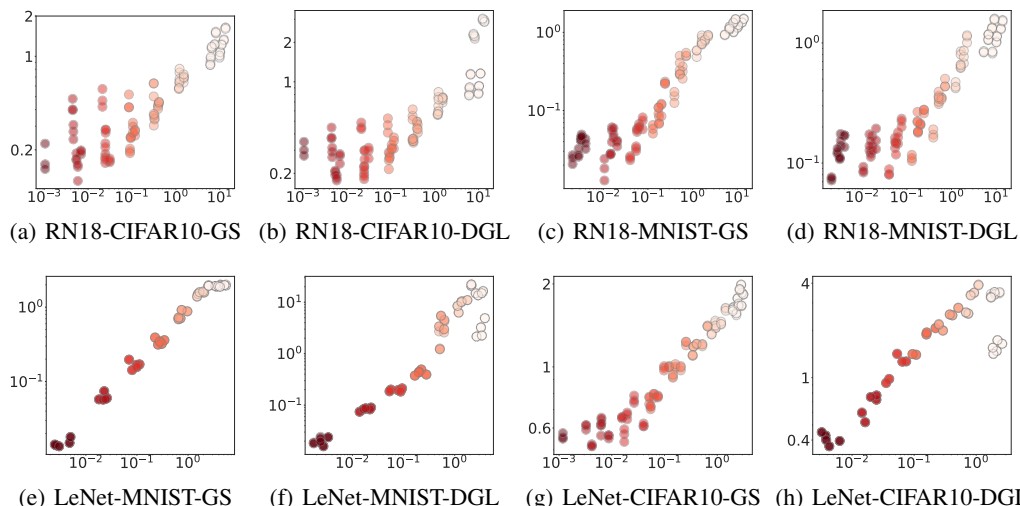

Figure 16: RMSE (y-axis) vs. $\mathcal{I}_{lb}$ (x-axis) with gradient pruning. A darker color indicates a smaller pruning ratio. It shows that $\mathcal{I}_{lb}$ is a good estimator of RMSE.

structural similarity and semantic distance between the original and recovered images, respectively. The relationship between SSIM and LPIPS is shown in Fig. 17. Since $\mathcal{I}_{lb}$ aims to lower bound the privacy risk in terms of RMSE, $\mathcal{I}_{lb}$ does not have the same scale as SSIM and LPIPS. Nevertheless, $\mathcal{I}_{lb}$ also has a positive correlation between SSIM and LPIPS, which implies that it is a good estimator for the structural similarity and semantic distance between the original and recovered images.

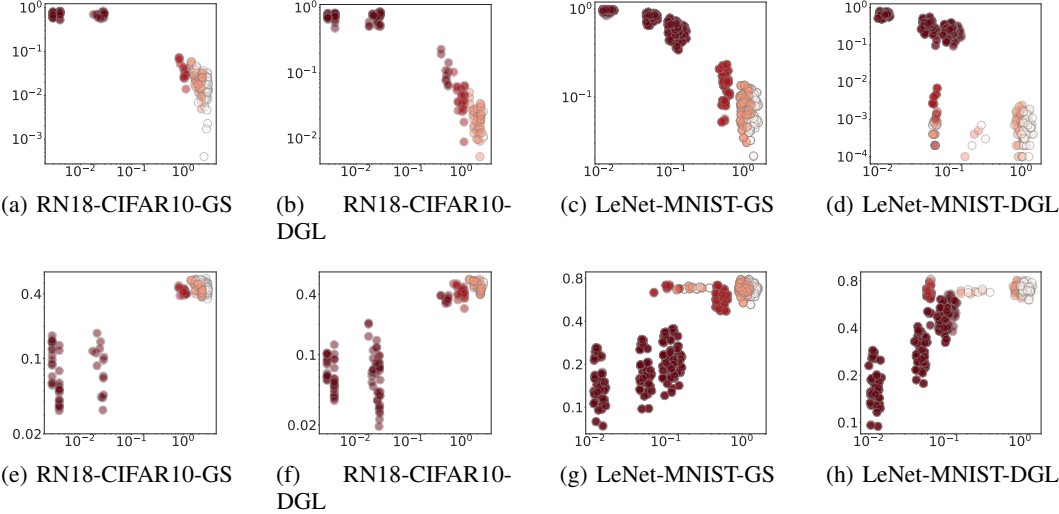

(a) RN18-CIFAR10-GS  (b) RN18-CIFAR10-DGL  (c) LeNet-MNIST-GS  (d) LeNet-MNIST-DGL

(e) RN18-CIFAR10-GS  (f) RN18-CIFAR10-DGL  (g) LeNet-MNIST-GS  (h) LeNet-MNIST-DGL

Figure 17: $\mathcal{I}_{lb}$ is positively correlated with these two metrics and is a good estimator for the structural similarity and semantic distance between the original and recovered images. Darker color indicates higher variance. **Top ((a)-(d)):** SSIM (y-axis) vs. $\mathcal{I}_{lb}$ (x-axis). **Bottom ((e)-(h)):** LPIPS (y-axis) vs. $\mathcal{I}_{lb}$ (x-axis). A higher SSIM and a lower LPIPS indicate a higher privacy risk.

## F   Dynamics of $\mathcal{I}_{lb}$ During Training

Previous existing empirical results show that privacy risk decreases by training epochs (Balunović et al., 2022; Geiping et al., 2020). We evaluate the dynamics of $\mathcal{I}_{lb}$, RMSE with DGL attack and RMSE during training in Fig. 18. It shows that as the training epoch increases, the $\mathcal{I}_{lb}$ also has an increasing trend. While the RMSE of RN18 on the CIFAR10 dataset has a similar trend as $\mathcal{I}_{lb}$, that of LeNet on the MNIST dataset has a significant rise at the epoch 60, which is due to the slower learning speed of LeNet than RN18. Moreover, almost for all the epochs, there is a sample with low $\mathcal{I}_{lb}$, which again emphasizes the unfairness in privacy protection.

## G   The Impact of $\epsilon$ on Efficient Matrix Inversion

In Fig. 19, we study the impact of $\epsilon$ on efficient matrix inversion proposed in Section 4. We evaluate the impact on the LeNet with the MNIST dataset. The y-axis is the RMSE. $\mathcal{I}$ (matrix inversion) is calculated as defined in Eq. (4). $\mathcal{I}_{lb}$ (matrix norm) is calculated as defined in Eq. (5). It shows with $\epsilon \in [1, 10]$, $\mathcal{I}_{lb}$ is a lower bound of the RMSE. It also shows that $\mathcal{I}_{lb}$ is an accurate estimator of I$^2$F. Thus, we can directly use $\mathcal{I}_{lb}$ in practice to lower bound the privacy risk to avoid fine-tuning the $\epsilon$.

## H   Discussion

### H.1   Validity of Assumption 3.4-3.5

We make assumptions about the Lipschitz continuous Jacobian and gradient in Assumption 3.4 and Assumption 3.5, respectively. These two assumptions are not necessary for I$^2$F but are only used to provide a theoretical validation of I$^2$F when the noise $\delta$ is not infinitesimal. To discuss the validity of these two assumptions in practice, we calculate the value of $\mu_J$ and $\mu_L$ of LeNet with two datasets.

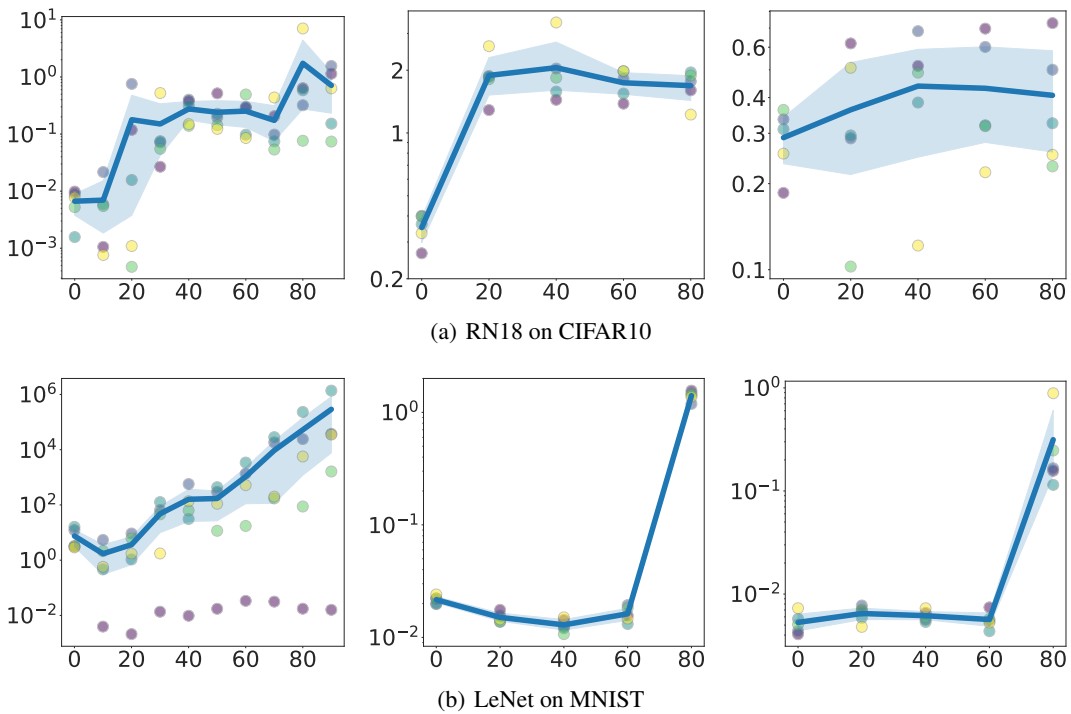

(a) RN18 on CIFAR10

(b) LeNet on MNIST

Figure 18: Privacy risks decrease by training epochs ($x$-axis). Different colors indicate different samples. The y-axis from left to right: $\mathcal{I}_{lb}$, RMSE w/ DGL attack, RMSE w/ GS attack whose smaller values indicate lower risks. The blue line indicates the mean value and the shadow is the variance (some outliers are dropped). The noise is sampled from a Gaussian distribution with a mean zero and variance $10^{-3}$.

For the CIFAR10 dataset, $\mu_L = 0.5014$ and $\mu_J = 1.7 \times 10^{-13}$. For the MNIST dataset, $\mu_L = 0.7192$ and $\mu_J = 3.7 \times 10^{-13}$. These values are not so large that they are reasonable in practice.

## H.2  Extension of I²F to the GS Attack

The derivation of I²F is considered in the DGL attack as defined in Eq. (1), but I²F can also be applied to the GS attack. Note that the minimizer of the DGL attack is one solution to the GS attack. That means the GS attack can be attained by an optimal DGL attack which is our assumption. Therefore, the DGL attack-based theorem is applicable to the GS attack.

Empirically, we evaluate $\mathcal{I}_{lb}$ on GS attack in Figs. 2 and 16. It shows that $\mathcal{I}_{lb}$ is linearly correlated to the metrics of RMSE, which proves the utility of I²F under GS attack.

## H.3  Extension of I²F with Prior Knowledge

Our theorem of I²F can be extended to take into account the prior knowledge. Consider the inversion optimization problem with prior knowledge as $\min_x L'_I(x; g) = L_I(x; g) + I_C(x)$ where $I_C(x)$ constrains $x$ in the prior space $C$ and $L_I(x; g) = \|\nabla_\theta L(x; \theta) - g\|$ defined in Eq. (1). Then the optimization problem can be rewritten as $\min_{x \in C} L_I(x; g)$. Thus, as long as the original image $x_0$ is in the feasible region defined by $I_C(x)$, our Assumption 3.1 and theorems are also applicable. Intuitively, a good regularization should satisfy the requirement, otherwise, it will unreasonably reject the correct recovery.

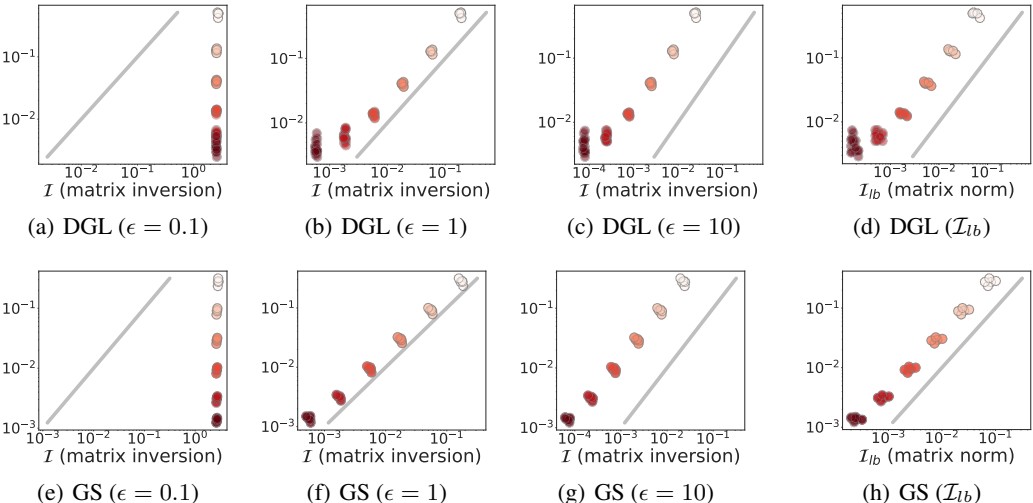

Figure 19: The impact of the value of $\epsilon$ is evaluated on LeNet with the MNIST dataset. The y-axis is the RMSE. (a)(b)(c)(d): DGL attack. (e)(f)(g)(h): GS attack. It is observed that $\mathcal{I}$ (matrix inversion) is effective with $\epsilon \in [1, 10]$ but not $\epsilon = 0.1$. It shows that (1) there exists a range of $\epsilon$ where $\mathcal{I}_{lb}$ can lower bound the RMSE; (2) $\mathcal{I}_{lb}$ is an accurate estimator of I²F, thus we can avoid fine-tuning $\epsilon$.

## H.4 Discussion with Prior Works

Closest to our work, Hannun et al. (2021) provided a second-order worst-case metric for analyzing privacy attacks. However, our work provides novel contributions both on technique and implications which essentially root from the proposed I²F metric.

**Technical difference.** First, we focus on a different scope against (Hannun et al., 2021). Hannun et al. (2021) proposed Fisher information loss (FIL) to measure the information leakage risk in the context of model inversion and attribute inference, e.g., only the attribute inference is considered in their experiments. Instead, we evaluate the privacy risk under gradient inversion attacks. Second, our metric is more scalable and applicable to large models. For example, in Eq. (18) in (Hannun et al., 2021), the inverse of the Hessian matrix needs to be calculated even when quantifying the information leakage of only one sample, which is inefficient and intractable for large models. Because of the computation inefficiency, only linear regression and logistic regression models are considered in their theories and experiments. Instead, we verify the feasibility of I²F in much larger models like ResNet152 on ImageNet in Fig. 3.

**Our new findings.** First, though the unfairness of information leakage of different samples was discussed in (Hannun et al., 2021), we investigate the issue in a different attack method and justify the commonness of the unfairness in different attacks. Second, we additionally provide other insights than (Hannun et al., 2021). For instance, the influence on gradient inversion of different initialization methods is studied. We also find the influence of perturbations is not equivalent even in the same noise scale. We believe these insights are also critical to the privacy and security community, especially in the area of gradient inversion.

Besides, (Guo et al., 2022; Hayes et al., 2023) propose to bound the reconstruction attack in terms of the attack success rate and the expectation of the $L_2$ distance between the recovered and original image. Nevertheless, their conclusions are based on the differential privacy (DP) quantification framework so it is hard to analyze the influence of other defense mechanisms such as gradient pruning and arbitrary noise. Also, they bound the privacy risk from the statistical sense raised by the randomness of DP, while our work can evaluate the sample-wise worst-case privacy risk at any time during the model training. Moreover, they assume the access of the attacker to the remaining samples (except for the privacy sample) or the multi-round gradient, which is not practical in real gradient inversion scenarios.

### H.5 Discussions about the Worst-case Assumption

Our work is mainly built upon the Assumption 3.1 that there is only a unique minimizer for $L_I(x; g)$ given a gradient vector $g$. Because of the hardness of optimizing a non-linear objective in attack (Eq. (1)), the worst-case may not be reachable in practice. Here, we discuss when the assumption has to be relaxed and why our method is still applicable. In addition, we emphasize that a stronger attacker exists theoretically, resulting in the necessity of a worst-case assumption.

**Non-bijective inversion mapping** $G_r(g)$. $G_r(g)$ could be non-bijective when the loss function is non-convex. In other words, given the same $g$, the output of $G_r(g)$ could include multiple choices. We want to argue that this case does not conflict with our assumption. Consider an attack given the exact gradient of a sample. Note that the sample itself is a solution to Eq. (1). Thus, given the exact gradient of a sample, the attack can exactly recover the sample. Even if the solution is non-unique, we can still essentially assume the attack can attain the sample in the worst case.

**Optimizing the gradient inversion objective may not converge to the original image.** Note that the original image is always an optimal solution for the inversion loss in Eq. (1). Even though the convergence is not guaranteed, there always exists an algorithm that can converge to the original image. To our best knowledge, there is no evidence to show the attack cannot approach the worst case where the original input is recovered. Instead, empirical results have shown that the images can be recovered almost perfectly (Geiping et al., 2020). Thus, due to the sensitivity of privacy, a worst-case assumption is necessary to strictly bound possible privacy risks with arbitrary strong attacks, which is commonly imposed by the literature (Dwork, 2006; Abadi et al., 2016).

## I  Realistic Impact

Federated learning (FL) (McMahan et al., 2017) is a popular distributed training paradigm that benefits from the data and computation sources from multiple clients. As a principle of FL, clients will upload the local gradient based on the private data to the server for the concerns of data privacy and safety, instead of directly sharing the private data of each client. However, recent works (Geiping et al., 2020; Zhu et al., 2019) show that an attacker, who has access to the local gradient (e.g., a malicious server), can leverage the local gradient to recover the private data of the clients, which we call Deep Gradient Leakage (DGL). Even with large models like Transformer, the attacker can still successfully recover the private data given the gradient (Hatamizadeh et al., 2022).

Auditing potential privacy risks is essentially desired for privacy-sensitive applications, including but not limited to finance (Long et al., 2020), healthcare (Antunes et al., 2022; Xu et al., 2021), and clinical data (Dayan et al., 2021; Roth et al., 2020). Such privacy concerns have been discussed extensively in previous work. For instance, Zhang et al. (2023) raises the privacy concern of FL in financial crime detection while (Kaissis et al., 2021; Li et al., 2023) discusses it in the medical and healthcare applications, respectively.

To echo the demands for privacy risk auditing, we provide a fundamental tool for bounding the worst-case risks of DGL. We show multiple insights, for example, the unfairness of privacy protection using random noise defense. Thus, we expect our work can call the attention of the community to the privacy concern raised by DGL, especially the worst-case instance-level privacy risk. Moreover, We expect $I^2F$ to be a keystone for designing more powerful defense mechanisms.

