# OpenReview forum: "Understanding Deep Gradient Leakage via Inversion Influence Functions"
_NeurIPS.cc/2023/Conference — NeurIPS 2023 poster_

### Official Review · Reviewer_X287 · 2023-07-05

**Soundness:** 3 good
**Presentation:** 3 good
**Contribution:** 2 fair
**Rating:** 7
**Confidence:** 4

**Summary:**

**Key Contribution:** This work proposes a new method for analysis of privacy risk in the deep leakage from gradients attack, which does not rely on assumptions of the model architecture or attack optimization method.

**Approach:** The work defines an “Inverse Influence Function” (I2F) which is able to determine some metric of privacy risk given an applied defense to gradients, irrespective of the model architecture or attack optimization technique. The authors use their framework to identify that the eigenvalues of the matrix $JJ^T$ greatly impacts the MSE of any possible recoveries - in particular, the paper identifies that the smallest eigenvalues can greatly reduce the risks of leakage.

**Evaluation:** The paper evaluates their approach using LeNet and ResNet18, trained on MNIST and CIFAR10. They consider the impacts of different eigenvalues on the predicted MSE in their I2F framework, and find that the most risky samples (i.e. lowest MSE recoveries) tend to be those where the jacobian had large eigenvalues. The paper also evaluates the impacts of different model initialization schemes on recovery MSE.

**Strengths:**

**Originality:**
The paper reframes analysis of DLG attacks in terms of prior work by Koh and Liang. In this way, it is an original framing of the DLG problem, using established techniques.

**Quality:**
The paper is of good quality, providing interesting conclusions and analysis of the I2F approach.

**Clarity:**
The paper is well-written and clear - the mathematical steps are simple and easy to follow, with intuitive results.

**Significance:**
The paper builds a simple yet intuitive mathematical framework to estimate privacy risk in DLG attacks. The key significance here is in the simplicity and efficiency of the approach, which allows it to be easily used in practice.

**Weaknesses:**

**Originality:** The paper claims that prior work by  Fan et al. (2020) is not generalizable to different architectures, such as convolutional networks. However, in Fan et al. (2020) Appendix A it is established that such layers can be re-written as fully-connected ones. It is unclear if the paper’s claims about the limitations of prior approaches are true. A more in-depth comparison with prior would help establish the novelty of the paper.

**Limited Evaluation:** The scope of the evaluation could be broadened to further strengthen the paper’s conclusions. It would be interesting to see how the estimation of MSE changes with respect to training iterations - prior work (as cited in the paper) demonstrates that trained models are more susceptible to the DLG attack, and hence justifying this empirical finding with the I2F framework would lend more credence to its utility. Moreover, the paper only considers two defenses: gradients perturbation, and mixing of training samples. However, the paper could be stronger if it considered more defenses e.g. gradient pruning, which is a commonly considered defense.

**Limited Applicability to Realistic Scenarios:** The paper cites Huang et al. 2021 and mentions that BN statistics are needed for a realistic inversion. However, it is unclear how the lack of BN statistics impacts the I2F framework’s utility in a realistic scenario.

**Questions:**

- (40) What does “converging to the optimal attack” mean in this sentence? Do you mean a perfect recovery of the training data? Or that the optimization method used in the attack itself is optimal?

- (145) Only considering MSE recovery leaves out situations where the recovered image is semantically the same as the private image, but differs in individual pixels. Have you considered alternative metrics which are less sensitive to individual pixels e.g. total variation?


**Minor nits:**
The original attack is referred to as  “Deep Leakage from Gradients” - is there a reason you chose a different terminology in your paper?

**Limitations:**

The authors sufficiently addressed the potential social impact and lmiitations of their work.

---

> ### Author Rebuttal · Authors · 2023-08-10
>
> Thanks a lot for the positive views of the quality and significance of our work! We are glad to address your concerns.
>
> **W1: (Originality)** Fan et al. (2020) can generalize to other architectures, like CNNs.
>
> **A1:** Thanks for the question.
> - Fan et al. (2020) mentioned that their method can be generalized to convolutional layers. We apologize for the mistake and will revise it in the final paper.
> - The main difference between our work and Fan et al. (2020): Fan et al. (2020) provided a good understanding of when the initial input data be determined and propose an **upper bound of the relative error**, which measures the best-case privacy risk considering the weakest attack. However, our work assumes there exists a stronger attack and focuses on the **lower bound of the RMSE**. Thus, what we consider is the worst-case privacy risk. Both the upper and lower bound are important to better understanding towards gradient inversion attack. So we believe **Fan et al. (2020) and our work are both critical to gradient inversion attack**.
> - **Moreover**, other privacy quantifications like differential privacy [E, F] also measure the worst-case privacy risk, so we believe our target following prior work on the worst case is reasonable.
>
> **W2 (Limited Evaluation):** How does I2F change in training? Evaluate I2F on grad pruning.
>
> **A2:** Thanks for the suggestion.
> - **I2F during training:** In Fig.4 of the attached PDF, we evaluate the change of $I_{lb}$ and recovery MSE during model training. It is observed that **$I_{lb}$ increases by epochs, which means a decreasing privacy risk**. This is consistent with the previous existing empirical results that a well-trained model is more difficult to be inverted by gradient than a randomly initialized model [B, C] (as we claimed in lines 290-292).
> - **I2F with gradient pruning:** In Fig.1 of the attached PDF, we evaluate I2F under the gradient pruning defense where I2F is linearly correlated with RMSE. Thus, **I$^2$F is generalizable with gradient pruning defense**.
>
> We will add this interesting discussion to the final paper.
>
> **W3 (Limited Applicability):** Based on Huang et al. (2021), how does the lack of BN impact the I2F framework’s utility
>
> **A3:** Thanks for the comments.
> - First, we are at a different stand than Huang et al. (2021). Huang et al. aim to strengthen the attack by leveraging the BN statistics which was shown to be more effective than attacks without BN statistics. In other words, **lacking BN statistics as regularization may weaken the attacks**.
> - In contrast, we aim to estimate the risks from potential gradient inversion attacks from the victim's stand. The victim, who provides the gradient, always has access to the full network including BN statistics. **Without BN statistics, we may underestimate the risks and therefore leave our victim at stake**.
> - Unlike Huang et al., BN statistics are not explicitly used for regularizing and improving inversion attacks in I2F. With the same I2F formulation, **we can estimate the risks from both vanilla gradient inversion or the one with BN-guided regularization (Huang et al., 2021)**.
>
> **Q1:** What does “converging to the optimal attack” mean (line 40)
>
> **A1:** We assume the attack is perfect given the exact gradient of a sample.
> - Note that the sample itself is a solution for Eq(1). If the solution of the inversion attack is unique, then it is equivalent to saying the sample is the unique optimal solution for solving Eq (1). Thus, given the exact gradient of a sample, the attack can exactly recover the sample.
> - Even if the solution is not unique, we can still essentially assume the attack can attain the sample in the worst case.
> - When noise exists, it is unlikely to recover the original images, but we assume the attack can recover the image that is closest to the original image. We will revise the paper to make it more clear.
>
> **Q2:** MSE lacks semantic info. Try other metrics that are less sensitive to individual pixels.
>
> **A2:** Thanks for the suggestion.
> - We follow most existing work [B, C, D] to use MSE (or PSNR, which is negatively correlated with MSE) as the default measure but we agree on the insufficiency of MSE for larger and more complex models.
> - In Fig.3 of the attached PDF, we try two metrics that capture more structural and semantic information between images, i.e., SSIM and LPIPS[A]. SSIM measures the structural similarity between two images, and LPIPS measures the semantic distance. We consider RN18 on CIFAR10 and LeNet on MNIST both with GS and DGL attacks. We find that **$I_{lb}$ is linearly correlated with these two metrics and is a good estimator for the structural similarity and semantic distance between the original and recovered images**.
>
> **Minor:** Why use DGL instead of DLG (Deep leakage from gradients)
>
> **A:** We used DGL because that “Deep Gradient Leakage” is more concise than “Deep Leakage from Gradients”. We will revise in final paper to avoid confusion.
>
> [A] Zhang, Richard, et al. "The unreasonable effectiveness of deep features as a perceptual metric." CVPR. 2018.
>
> [B] Balunović, Mislav, et al. "Bayesian framework for gradient leakage." arXiv (2021).
>
> [C] Geiping, Jonas, et al. "Inverting gradients-how easy is it to break privacy in federated learning?." NeurIPS (2020).
>
> [D] Zhu, Ligeng, Zhijian Liu, and Song Han. "Deep leakage from gradients." NeurIPS (2019).
>
> [E] Dwork, Cynthia. "Differential privacy." International colloquium on automata, languages, and programming. Berlin, Heidelberg: Springer Berlin Heidelberg, 2006.
>
> [F] Abadi, Martin, et al. "Deep learning with differential privacy." ACM SIGSAC. 2016.

---

> ### Author Response · Authors · 2023-08-20
> **Follow-up on rebuttal and a kind reminder**
>
> Dear Reviewer X287,
>
> We want to thank you for your constructive suggestions and thoughtful reviews, which are valuable to improving our paper. As a follow-up on our rebuttal, we would like to kindly remind you that the close date of the discussion is approaching. We hope to use this open response window to discuss the paper, answer follow-up questions, and improve the quality of our paper. Have you gotten a chance to read our rebuttal, in which we tried our best to address your concerns? We want to make sure that you found our responses solid and convincing. And we would be more than happy to provide more information or clarification.
>
> Authors

---

> > ### Comment · Reviewer_X287 · 2023-08-20
> >
> > Thank you for the thorough responses to my concerns, and the additional follow-up reminder.
> >
> > I am satisfied with the author's responses to my questions, and have raised my score of the paper. My only suggestion is that the authors include more detail about the scenario related to batchnorm statistics, as prior DGL attacks assume access to these statistics and include them as a regularization term.

---

> > > ### Author Response · Authors · 2023-08-21
> > > **Thanks for raising your score**
> > >
> > > We are glad that all your concerns are addressed. We really appreciate your valuable comments and acknowledging the contribution of our work! Based on your suggestion, we will revise our final paper to include a more detailed discussion on BN prior.

---

### Official Review · Reviewer_VRVE · 2023-07-05

**Soundness:** 3 good
**Presentation:** 3 good
**Contribution:** 3 good
**Rating:** 6
**Confidence:** 4

**Summary:**

This paper proposes inverse influence function (IIF), the indicator of how reconstructed input (from gradient inversion attack) changes with respect to a gradient change. This function can be simply formulated using Jacobian and gradient change. The correlation between the proposed measure and reconstruction quality is demonstrated both theoretically and empirically. Also, IIF can predict the vulnerability of gaussian perturbation (probably for privacy protection) based on Jacobian (or sample). Using this fact, the sample whose Jacobian has larger maximum eigenvalue is considered to be still unsafe from perturbation-based protection. This fact is also empirically demonstrated on MNIST. Various model initialization schemes are also analyzed using eigenvalues of Jacobian.

**Strengths:**

1. The paper is easy to follow.
2. The formulation of IIF is novel and interesting.
3. The correlation between the proposed measure and reconstruction quality is empirically demonstrated on several datasets
4. Both theoretical and experimental analyses on Gaussian perturbation based privacy protection using IIF is novel and important.

**Weaknesses:**

1. There is no experimental result on large-scale dataset and large DNNs.
2. The worst-case assumption can be very strong in some cases when optimal point from the attack does not reach ground truth.
3. The theory is strongly based on L2 distance between gradients. How about GS attack case, which is based on cosine-similarity loss?

**Questions:**

Please see the weaknesses for basic questions.
An another question: what is the conclusion from several analyses on model initialization or perturbation-based protection using IIF?

**Limitations:**

Limitation is included in the submission.

---

> ### Author Rebuttal · Authors · 2023-08-10
>
> We really appreciate the valuable comments from the reviewer. We are glad to address the concerns.
>
> **W1:** No experiments on large DNNs and large datasets
>
> **A1:** Thanks for the suggestion. In Fig.5 of the attached PDF, we evaluate our metric on the large model (ResNet152) and large dataset (ImageNet).
> 1. For larger models, the MSE is no longer a good metric for the recovery evaluation. Even if state-of-the-art attacks are used and the recovered image is visually similar to the original image in Fig.5(b), the two images are measured to be different by MSE, due to the visual shift: The dog head is shifted toward the left side. To capture such shifted similarity, we use LPIPS [C] instead, which measures the semantic distance between two images instead of the pixel-to-pixel distance like MSE.
> 2. Fig.5(a) shows that I2F is correlated to LPIPS using large models and image scales. This implies that **I2F is a good estimator of recovery similarity**.
>
> **W2:** The worst-case assumption can be very strong in some cases when the optimal point from the attack does not reach ground truth.
>
> **A2:** Thanks for the comments.
> - It is easy to see that the ground truth (the private image) is one optimal solution for the attack objective. The problem may be if the optimization algorithm can converge to the desired optimal solution. If this is the problem, we believe the attack can evolve to be much stronger in the near future. We have already seen many recent attack approaches that are getting increasingly powerful [D, E]. Thus, it is important to bound the risk in advance before the risk becomes true.
> - **In addition**, worst-case risk estimation is a common practice. For example, differential privacy [A, B] bound the worst-case chance that the sample is identifiable.
>
> **W3:** Theory is based on L2-norm attack. How about the cos-sim attack?
>
> **A3:** Thanks for this question. **Our metric is also applicable for evaluating cos-sim attacks**.
> - Note that the minimizer of the L2-norm attack is one solution to the cos-sim (GS) attack. That means **the cos-sim attack can be attained by an optimal L2-norm attack which is our assumption**. Therefore, the L2-norm-based theorem is applicable to the cos-sim attack, as well.
> - **Empirically**, we evaluate $I_{lb}$ on GS attack based on cosine-similarity inversion loss as in Fig.2 and Fig.12 of the main body and appendix of our paper, respectively. It shows that **$I_{lb}$ is linearly correlated to the metrics of MSE, which proves the utility of our metric under GS attack**. We will include the discussion in the revision.
>
> **Q1:** What is the conclusion from several analyses on model initialization or perturbation-based protection using IIF?
>
> **A1:**  Thanks for the question.
> 1. Firstly, we find that the **unfairness of privacy protection exists in both samples and classes**. Even for the same noise (distribution), some samples or classes may be more vulnerable to leak privacy as shown in **Fig.5**.
> 2. Secondly, **the protection effect of the perturbation itself is not fair**, which is implied from the results that the eigenvectors with smaller eigenvalues are more effective on protection as shown in **Fig.4**, even though all the eigenvectors have the same norm.
> 3. Thirdly, since a well-trained model is much more difficult to be inverted with gradient, we evaluate the effect of different initializations on inversion and find that **Kaiming and Xavier have less privacy risk than the normal and uniform initializations** as shown in **Fig.6**. We will include a more clear conclusion in the revision.
>
> [A] Dwork, Cynthia. "Differential privacy." International colloquium on automata, languages, and programming. Berlin, Heidelberg: Springer Berlin Heidelberg, 2006.
>
> [B] Abadi, Martin, et al. "Deep learning with differential privacy." ACM SIGSAC. 2016.
>
> [C] Zhang, Richard, et al. "The unreasonable effectiveness of deep features as a perceptual metric." CVPR. 2018.
>
> [D] Haim, Niv, et al. "Reconstructing training data from trained neural networks." NeurIPS (2022)
>
> [E] Kariyappa, Sanjay, et al. "Cocktail party attack: Breaking aggregation-based privacy in federated learning using independent component analysis." ICML, 2023.

---

> > ### Comment · Reviewer_VRVE · 2023-08-17
> > **Response to the Rebuttal**
> >
> > Thanks for the rebuttal.
> >
> > Response to A1 : Did you apply cos-sim attack in large DNN experiments? As far as I know, the third dog image can be recovered well using cos-sim attack in the paper "Inverting Gradients -- How easy is it to break privacy in federated learning?" (Geiping et al.). Are hyperparameters are tuned enough?
> >
> > Response to A2 : The original purpose of this work is to measure the "change" in reconstruction quality from gradient manipulation. I appreciate that sometimes it is hard to derive the formula without such an assumption. Nevertheless, the application of I^2F seems questionable when input cannot be recovered perfectly from gradient. Can we focus only on "change" without such assumption?
> >
> > Response to A3 : Unfortunately, I cannot agree with the assumption that cos-sim attack is the optimal L2-norm attack.
> >
> > Response to (A to Q1) : Thanks for addressing the question.

---

> > > ### Author Response · Authors · 2023-08-19
> > > **Response to Reviewer VRVE**
> > >
> > > Thanks for your response. We are glad to address your concerns.
> > >
> > > - **R1:** Yes, we apply cos-sim attack following [F] (Geiping et al.) in the large DNN experiments. We follow their hyperparameters and tuned them in our case to ensure a high-quality reconstruction.
> > >     1. Though the original dog image is the same in the two experiments, our experiment setting is different from and is harder than the one of the mentioned dog image Fig.1 in [F]. Our experiment is conducted on a hard setting with noised gradient instead of raw gradients. With a clean gradient, the visual results of the dog image from [F] and our experiments are very similar.
> > >     2. We have carefully tuned the hyperparameters and **our results are consistent with those in [F]**. In Fig.13 and 14 in [F], the visual results of recovered images for RN152 on ImageNet are also worse than those of RN18. Meanwhile, both [F] presents similar visual shift as our results in the recovered images.
> > >     3. Worth to mention, the large DNN experiments aim to show the relationship between our metric and LPIPS with a large model and dataset. Fig.5 in the attached PDF already shows that **I2F is a good estimator of recovery similarity**.
> > >
> > > - **R2:** Thanks for your follow-up questions.
> > >     - First, we argue that the worst-case assumption is not proposed for the ease of derivation but is essential and common in practice.
> > >         1. The original image is always an optimal solution for the inversion loss in Eq.1., therefore **theoretically, there always exists an algorithm that can converge to the original image**.
> > >         2. To our best knowledge, there is no evidence to show the attack cannot approach the worst case where the original input is recovered. Instead, empirical results have shown that the images can be recovered almost perfectly [F] (Geiping et al.). Thus, due to the sensitivity of privacy, a worst-case assumption is necessary to strictly bound possible privacy risks with arbitrary strong attacks, which is commonly imposed by the literature [A, B, G].
> > >     - Second, it is an open yet interesting question to investigate the change without the worst-case assumption. However, we argue that such analysis has the following difficulty:
> > >         - To analyze the change of reconstruction, we need to know the recovery with the clean gradient. Nevertheless, without the worst-case assumption, it is hard to quantify the relationship between the original and recovered images. We will further discuss this as a future direction in our final paper.
> > >
> > > - **R3:** We agree that the cos-sim attack is not the optimal L2-norm attack, which is **NOT** our claim, either. We believe there is a misunderstanding regarding our explanation of the relationship between the L2-norm and cos-sim attack. We argue that an optimal L2-norm attack solution is also an optimal one to the cos-sim attack, but not vice versa, because the cos-sim attack only considers the angle between the gradients of the original and recovered images. In other words, the L2-norm-based conclusion should apply to cos-sim attacks, (but not vice versa).
> > >
> > > [A] Dwork, Cynthia. "Differential privacy." International colloquium on automata, languages, and programming. Berlin, Heidelberg: Springer Berlin Heidelberg, 2006.
> > >
> > > [B] Abadi, Martin, et al. "Deep learning with differential privacy." ACM SIGSAC. 2016.
> > >
> > > [F] Geiping, Jonas, et al. "Inverting gradients-how easy is it to break privacy in federated learning?." NeurIPS 2020.
> > >
> > > [G] Guo, Chuan, et al. "Bounding training data reconstruction in private (deep) learning." ICML, 2022

---

> > > > ### Comment · Reviewer_VRVE · 2023-08-21
> > > >
> > > > Thank you for through responses to my questions.
> > > >
> > > > Regarding R1, there should be clarification about the noise to the gradient.
> > > > I thought that the result is based on clean gradient without the noise.
> > > > If the dog image is reconstructed from noisy gradient, the result has no problem.
> > > >
> > > > Regarding R2, I appreciate the work but I want the authors to add some comments on the the gap between theory and practice. For example, some calibration methods can be designed to consider the case that the image reconstructed from clean gradient is far from the ground truth.
> > > >
> > > > Regarding R3, The responses mitigate my concerns. However, I believe that IIF can be reformulated only for cos-sim attack.
> > > >
> > > > By the way, I appreciate the importance of the work in this field and large scale experiment is included. I would raise my score to 'Weak Accept'.

---

> > > > > ### Author Response · Authors · 2023-08-21
> > > > > **Thank you for agreeing to raise your score to ‘weak accept’**
> > > > >
> > > > > We are glad that most of your concerns are addressed. We really appreciate your valuable comments and raising your score! We will revise our final paper accordingly based on your comments. Just a kind reminder that your score has not been updated yet.

---

### Official Review · Reviewer_FFT2 · 2023-07-09

**Soundness:** 3 good
**Presentation:** 2 fair
**Contribution:** 3 good
**Rating:** 7
**Confidence:** 4

**Summary:**

The authors propose Inversion Influence Function ($I^2F$), a closed-form lower-bound approximation that estimates the recovery $L_2$-norm caused by gradient perturbation in gradient inversion attacks. Detailed mathematical proof and experiments are provided, with comparisons of privacy vulnerability with regard to data, model, perturbation, and attack methods.

**Strengths:**

- The theorem is supported with detailed math proof and experiment results.
- $I^2F$ gives a good approximation of privacy vulnerability with regard to data, model, perturbation, and attack methods. Most importantly, one can use it to find a theoretically optimal direction to perturb the gradient in federated learning.

**Weaknesses:**

- In Lemma B.3. (appendix), $\nabla_x L_I(x;g) = \nabla_x \|\| \nabla_\theta L(x, \theta) - g \|\|^2 = 2 \nabla_x \nabla_\theta L(x, \theta) (\nabla_\theta L(x, \theta) - g)$, the factor $2$ is missing.
- For figure 7 in the appendix, there's no explanation of what different lines mean, and it makes no sense to compare the running time of power iteration to that of inversion attack.
- There should be quoted conclusions/studies on the validity of assumptions, especially the actual value of $\mu_L$ and $\mu_J$. If they just exist but take very large values, Eq. (7) will be nonsense.
- $I_{lb}$ in Fig. (1) is an approximation of Eq. (4) and is different from  $I_{lb}$ defined in Eq. (5), and the computation speedup described in "Efficient Evaluation" is not used (only the convergence speed is evaluated).
- $L_2$-norm is used in almost all math equations, but the experiments use RMSE/MSE, which makes it hard to tell how tight the approximated lower bound is.
- In Sec 5.3, there exist both $\sigma(\theta^Tx)-1$ and $\sigma(\theta^Tx)-b$, which looks like an incomplete replacement.

**Questions:**

- What is the perturbation scale used in Fig. (4)?
- In Fig. (5), are gradient perturbations sampled from Gaussian noise, or do the follow the direction of the eigenvector with the smallest eigenvalue?

**Limitations:**

The proposed $I^2F$ relies on 5 assumptions that may not hold true. Out of these assumptions, 3.2  is satisfied by common loss functions, 3.3 is bypassed by "Extension to singular Jacobians". However, other assumptions, as well as the variables they introduce ($\mu_J$ and $\mu_L$), have not been well discussed.

> After rebuttal

We found the proposed method partially overlaps with [the Fisher Information Loss](https://arxiv.org/pdf/2102.11673.pdf), which weakens its technical novelty. However, there are still differences between the two works in terms of the lower bounds and algorithms they arrive at. I agree with AC that this work could be accepted conditioned on that a more thorough discussion of prior works is included in its final version.

---

> ### Author Rebuttal · Authors · 2023-08-10
>
> We really appreciate the affirmation of our contribution.
>
> **W1:** In Lemma B.3, factor 2 is missing
>
> **A1:** Thanks for pointing this out. We will revise it accordingly.
>
> **W2:** The explanation of Fig.7 and why comparing the convergence of power iteration and the attack.
>
> **A2:**
> - **What do different lines mean in Fig.7:**
>     - In Fig.7(a), different lines mean the convergence of power iteration under different random seeds.
>     - In Fig.7(b), different lines mean the convergence of the inversion loss under different learning rates (1, 0.5, 0.1, 0.05, 0.01). For each line in (b), it is repeated with 5 different random seeds.
> - **Why compare the running time of power iteration with the attack:**
>     - Conducting the inversion attack can be time-consuming and computationally intensive. **I2F is an attack-free analysis tool that can be used to evaluate the privacy risk given a sample and a perturbation. The efficiency gap between calculating I2F and conducting the inversion attack is a critical metric of the utility of I2F in practice.** Since we use $\frac{||J\delta||}{\lambda_{\max}(JJ^T)}$ to measure worst-case privacy risk, the computation cost exists in the numerator and denominator, respectively.
>     - The cost of the numerator can be calculated with a Jacobian-vector product, which is equivalent to two times of gradient computation with a vector production. So the numerator can be calculated efficiently with frameworks like PyTorch and its cost is constant for a given model and dataset.
>     - Thus, **the main cost comes from the denominator, which uses power iteration to calculate the maximum eigenvalue of $JJ^T$**. In Fig.7, we show the convergence of power iteration is about 20 times faster than that of the inversion loss, which means that our metric can evaluate the privacy risk accurately and more efficiently.
>
> **W3:** The validity of assumptions, especially the actual values of u_L and u_J. If they are large, then Eq7 makes no sense.
>
> **A3:** Thanks for the comments.
> 1. Assumption 3.1 is the assumption of a perfect attack, which is considered the worst-case privacy risk as we claim in lines 133-135. Worst-case risk measurement is currently used in many other areas, such as differential privacy [C]. As gradient inversion attacks are evolving stronger over time, we believe this assumption is reasonable and makes our measurement useful in the future.
> 2. Assumption 3.2 is satisfied with the common loss function like cross-entropy loss and was used in the literature [A, B].
> 3. For assumption 3.3, first, we agree that “this assumption is bypassed by the ‘Extension to singular Jacobians’” as shown in the limitations by the reviewer. Moreover, we show in Fig.1 and 2 that the lower bound of $I_{lb}$ is a good estimator of I2F. Thus we can directly use $I_{lb}$ to estimate worst-case privacy risk even when $JJ^T$ is singular.
> 4. Assumptions 3.4 and 3.5 is not necessary for I2F because I2F only depends on assumptions 3.1-3.3 (lines 151-152). Assumptions 3.4 and 3.5 are only used to provide a theoretical validation of I2F when the noise $\delta$ is not infinitesimal. We also calculate on LeNet the values of $\mu_J$ and $\mu_L$ in assumptions 3.4-3.5. **For CIFAR10, $\mu_L=0.5014$ and $\mu_J=1.7\times10^{-13}$. For MNIST, $\mu_L=0.7291$ and $\mu_J=3.7\times10^{-13}$.** These values are not so large that they are reasonable in practice.
>
> **W4:** I_{lb} in Fig1 is approx of Eq4, and is different from I_{lb} in Eq5. Computation speedup in “Efficient Evaluation” is not used.
> **A4:**
> - **$I_{lb}$ in Fig.1 is different from that in Eq.5:**
>     - We apologize for the confusion. The x-axis in Fig.1, i.e. *$I_{lb}$ (matrix norm)*, is calculated as defined in Eq.5. The matrix norm here is defined as $\|A\| = \sup_{x\ne0} (\|Ax\|/\|x\|)$ in line 122. The y-axis in Fig.1 is calculated as Eq.4, which is $(JJ^T)^{-1}J\delta$. We will clarify to avoid such confusion in the revision.
> - **The computation speedup in “Efficient Evaluation” is not used:**
> Three computation speedup has been proposed in “Efficient Evaluation”.
>     - Speedup (1) of efficient evaluation of $J\delta$ is used in the whole paper wherever we need to compute the Jacobian-vector product.
>     - Speedup (2), the efficient matrix inversion, is used in Fig.1, where we need to calculate $I$ (matrix inversion) as $(JJ^T)^{-1}J\delta$ in Eq.4. In Fig1., the results indicate that the lower bound $I_{lb}$ is a good estimator of $I$, so we use $I_{lb}$ instead in the following experiments.
>     - Speedup (3), the efficient evaluation of the Jacobian norm, is used in the whole paper wherever we need to compute the norm of the Jacobian.
>
>
> **W5:** The L2-norm is used in all math equations, but the experiments use MSE/RMSE, which makes it hard to tell how tight the approximated lower bound is.
>
> **A5:** We apologize for the confusion. There is a typo in Eq.1, in which the definition of recovery MSE should be the **root** of recovery MSE as $||x_0-G_r(g_0+\delta)||$. Also, please kindly check the caption of Fig.1 where RMSE is short for **root of mean square error** instead of *recovery MSE*. We will revise Eq.1 to make it more clear.
>
> **W6:** In Sec5.3, both 1 and b exist.
>
> **A6:** Thanks for pointing this out. There is a typo and it should be $\sigma(\theta^{T}x)-b$ instead of $\sigma(\theta^{T}x)-1$. We will fix this in the revision.
>
> **Q1:** What is the perturbation scale used in Fig4
>
> **A1:** In Fig.4, we use the eigenvector as the perturbation so the scale is 1.
>
> **Q2:** In Fig5, is the perturbation Gaussian noise or eigenvector?
>
> **A2:** The perturbation is sampled from Gaussian distribution as in the figure caption.
>
> [A] Guo, Chuan, et al. "Bounding training data reconstruction in private (deep) learning." ICML, 2022.
>
> [B] Hannun, Awni, Chuan Guo, and Laurens van der Maaten. "Measuring data leakage in machine-learning models with Fisher information." UAI, 2021.
>
> [C] Dwork, Cynthia. "Differential privacy." Springer Berlin Heidelberg, 2006.

---

> > ### Comment · Reviewer_FFT2 · 2023-08-21
> >
> > I appreciate the efforts the authors spent on their rebuttal! It solves my concerns and I've raised my score.

---

> > > ### Author Response · Authors · 2023-08-21
> > > **Thanks for raising your score**
> > >
> > > We are glad that all your concerns are addressed. We really appreciate your valuable comments and acknowledging the contribution of our work! We will revise our final paper based on your suggestions.

---

### Official Review · Reviewer_BBkr · 2023-07-12

**Soundness:** 3 good
**Presentation:** 2 fair
**Contribution:** 3 good
**Rating:** 6
**Confidence:** 3

**Summary:**

This paper proposes to leverage influence function as a tool for understanding and analyzing the privacy risk in gradient leakage by connecting the private gradients with the recovered images. Inversion Influence Function (I^2F) is introduced as an efficient approximation of deep leakage attacks. Theoretical justification is provided for this approximation and empirical analysis is conducted on two image datasets (MNIST and CIFAR10).

**Strengths:**

1. Understanding privacy leakage through gradients is an important and timely topic.
2. It is an original work and perhaps the first to leverage influence function to perform analysis on deep gradient leakage.
3. The analysis provides an explanation for sample- and class-wise variance in reconstruction quality (referred to as “unfair privacy” in the paper).


**Weaknesses:**

1. For complex neural networks with non-convex loss functions, the inversion mapping Gr may not be bijective, e.g., there might exist multiple x that induce a similar gradient.
2. The proposed inversion influence function based on first-order Taylor requires the added perturbation to be infinitesimal to be accurate. Although this is partially justified through Theorem 3.1, it can be seen from Fig. 1 that the lower bound is violated at larger noise values.
3. For singular Jacobians, a constant needs to be added for numerical stability, which has to be tuned for each specific dataset.
4. From what the reviewer understands, the performed analysis through influence function only considered an adversary who observes the (noisy) gradients and tries to perform inversion by solving the gradient matching optimization problem. In practice, a sophisticated attacker may leverage prior knowledge to improve the reconstruction. The experiments only involved attacks with no/weak prior (e.g., TV), it would be interesting to see how well the proposed metric approximates the reconstruction quality of a strong biased attacker, e.g., using GAN as prior.
5. In the batch recovery case, the upper bound by decomposing into individual gradient inversion might be too loose.
6. Not a weakness but some discussion and comparisons between the proposed lower bound and prior work on lower bounding reconstruction MSE would be nice to have, e.g., [R1] using estimation theory and [R2] using DP-SGD.
[R1] Guo, Chuan, et al. "Bounding training data reconstruction in private (deep) learning." International Conference on Machine Learning. PMLR, 2022.
[R2] Hayes, Jamie, Saeed Mahloujifar, and Borja Balle. "Bounding Training Data Reconstruction in DP-SGD." arXiv preprint arXiv:2302.07225 (2023).


**Questions:**

1. It can be seen from Fig. 2 that Cifar10 samples induce a larger variance compared to MNIST. Is this an artifact of Cifar10’s data distribution being more complex?
2. What are the variances of the Gaussian noise used to produce Fig. 1 & 2?
3. How much data is needed to tune the epsilon constant? How much impact does the constant have on the results? E.g., how would I^2F perform if an epsilon constant tuned on a different dataset is used?


**Limitations:**

The limitations and social impact have been discussed in the last section.

---

> ### Author Rebuttal · Authors · 2023-08-10
>
> Thanks for the positive comments on our contribution and novelty toward understanding the gradient inversion attack!
>
> **W1:** For complex NN and non-convex loss func, the inversion mapping $G_r$ may be not bijective
>
> **A2:** Thanks for your comments. Indeed $G_r$ may be not bijective but it is not a weakness of our work, since **it does not conflict with our assumption**. We assume a perfect attack given the exact gradient of a sample.
> - Note that the sample itself is a solution for Eq(1). If the solution of the inversion attack is unique, then it is equivalent to say the attack optimization method for solving Eq (1) is optimal. Note that the sample of the gradient is a solution for Eq(1). Thus, given the exact gradient of a sample, the attack can exactly recover the sample. Even if the solution is non-unique, we can still essentially assume the attack can attain the sample in the worst case.
>
> **W2:** I2F needs Talyor expansion, which requires perturbation to be infinitesimal.
>
> **A2**: Thanks for the question.
> 1. Firstly, with a small noise, it is more likely for the attack to recover an image closer to the original one [C]. Thus the privacy risk could be higher, so it is more valuable to bound it.
> 2. Second, since large noise can destroy the performance [C, D], a small noise is often a common and better choice so we focus more on the privacy risk under a smaller perturbation.
> 3. Third, we also provide a theoretical validation in Sec 3.2 to show the utility of $I^2F$ under non-infinitesimal noise, where we show I2F is still effective as a lower bound even with large noise.
>
> **W3&Q3:** For singular Jacobians, a constant $\epsilon$ should be tuned. How much impact does the constant have on the results?
>
> **A3:** Thanks for the question.
> 1. First, **we don’t need to fine-tune the constant $\epsilon$ in our experiments (and also in practice) except in Fig.1**. In Fig.1, it shows the extension for singular Jacobians (known as $I$ (matrix inversion)) can be estimated by $I_{lb}$. Thus we can use $I_{lb}$ to evaluate privacy risk without fine-tuning the $\epsilon$.
> 2. The impact of the constant $\epsilon$ is evaluated in Fig.2 of the attached PDF. There are two main observations.
>     1. First, there exists a range of $\epsilon$ where I$^2$F can be used to estimate the privacy risk accurately. So the target of fine-tuning $\epsilon$ is to find an optimal range instead of an optimal value, which makes fine-tuning much easier.
>     2. Second, in the optimal range of $\epsilon$, $I_{lb}$ is an accurate estimator of I$^2$F, thus the fine-tuning of $\epsilon$ can be avoided.
>
> **W4:** The experiments only involve attacks with no/weak prior (TV loss), more stronger prior should be used.
>
> **A4:** Thanks for the suggestion. We use the TV loss as a prior when attacking LeNet because the attack is strong enough. While when attacking RN18, we use BN statistics as a stronger prior as explained in lines 527-533. Fig.2 shows the evaluation of the effectiveness of I2F with GS attack with BN prior. It indicates that under a stronger attack with BN prior, I2F can still estimate the worst-case privacy risk.
>
> **W5:** Loose bound of batch recovery
>
> **A5:** Thanks for the question. We follow prior work like [E, F] and investigated the privacy risk of a single sample, and therefore the main focus is on the sample-wise attack. The upper bound in “Batch data” in lines 236-240 is an application of sample-wise attack analysis when the attacker can get the per-sample gradient in a batch instead of the average gradient of the batch. Under such a case, our proposed I2F can be applied to the worst-case batch-wise attack. We agree with the reviewer that a batch-wise extension of I2F is very useful and is an interesting future work.
>
> **W6:** Discussion with prior work.
>
> **A6:** Thanks for the suggestion.
> - [A] proposed semantic guarantees for DP mechanisms against training data reconstruction and two privacy accounting methods are evaluated based on their guarantees. The difference between [A] and our work lies in:
> 1. First, [A] discusses the model inversion attack, which is a weaker data reconstruction attack without gradient information compared to gradient inversion.
> 2. Second, [A] considers the bound of reconstruction error under the differential privacy framework, while our metric aims to measure the privacy risk under a general perturbation on the sample or gradient. Our work can evaluate the privacy risk under more defense mechanisms like gradient pruning or data perturbation.
> - [B] proposes an upper bound of the probability of the success of a reconstruction attack and shows that DP parameters are not sufficient to estimate the success of the attack. The difference between [B] and our work lies in:
>     - We provide a different risk measurement against [B]. [B] captures the privacy risk via the success probability of the reconstruction attack, while our work directly evaluates the reconstruction error under the worst case.
>
> We will include this discussion in the revision.
>
> **Q1:** Larger variance of CIFAR10 in Fig.2 is because .
>
> **A1:** Yes. We agree with you. Since CIFAR10 is more complex than MNIST, the variance of CIFAR10 is larger than that of MNIST.
>
> **Q2:** Noise variance in Fig1 and 2
>
> **A2:** It is $10^{-3}$, $10^{-4}$, $10^{-5}$, $10^{-6}$ from light to dark colors.
>
>
> [A] Guo, Chuan, et al. "Bounding training data reconstruction in private (deep) learning." ICML, 2022
>
> [B] Hayes Jamie et al. "Bounding Training Data Reconstruction in DP-SGD." arXiv 2023
>
> [C] Zhu, Ligeng et al. "Deep leakage from gradients." NeurIPS 2019
>
> [D] Huang, Yangsibo, et al. "Evaluating gradient inversion attacks and defenses in federated learning." NeurIPS 2021
>
> [E] Gao, Wei, et al. "Privacy-preserving collaborative learning with automatic transformation search." CVPR. 2021
>
> [F] Sun, J., et al. "Provable defense against privacy leakage in federated learning from representation perspective. arXiv 2020

---

> > ### Comment · Reviewer_BBkr · 2023-08-18
> > **Thank you**
> >
> > Dear authors,
> >
> > Thank you for your detailed response. Most of my concerns from the previous round have been addressed. Regarding W4, could you elaborate further on whether the proposed theoretical framework can be extended to take into account the adversary's prior knowledge?

---

> > > ### Author Response · Authors · 2023-08-19
> > > **Our theorem can be extended with prior knowledge**
> > >
> > > We are glad that most of your concerns are addressed.
> > >
> > > - **Response to W4:** Yes, our theorem can be extended to take into account the prior knowledge. Consider the inversion optimization problem with prior knowledge as $\min_x L’_I(x;g) = L_I(x;g) + I_C(x)$ where $I_C(x)$ constrains $x$ in the prior space $C$ and $L\_I(x;g) = ||\nabla\_{\theta}L(x;\theta) - g||$ defined in Eq.1.
> > > Then the optimization problem can be rewritten as $\min\_{x \in C} L_I(x;g)$. Thus, as long as the original image $x_0$ is in the feasible region defined by $I_C(x)$, our assumption 3.1 and theorems are also applicable. Intuitively, a good regularization should satisfy the requirement, otherwise, it will unreasonably reject the correct recovery.

---

> > > ### Author Response · Authors · 2023-08-21
> > > **Follow-up on rebuttal and a kind reminder**
> > >
> > > Many thanks for your valuable comments and response. As a follow-up on our rebuttal, we would like to kindly remind you that the close date of the discussion is approaching. Have you gotten a chance to read our responses above, in which we tried our best to address your concern? And we would be more than happy to provide more information or clarification.

---

> > > > ### Comment · Reviewer_BBkr · 2023-08-21
> > > > **Response to follow-up**
> > > >
> > > > Thank you for the clarification and your kind reminder. My questions have been addressed and I'm keeping my score in support of accepting the paper.

---

> > > > > ### Author Response · Authors · 2023-08-21
> > > > > **Thanks for your valuable comments**
> > > > >
> > > > > We are glad that all your concerns are addressed. Many thanks for reading our responses and lettering us know your thoughts. We really appreciate your valuable comments and will revise our final paper accordingly.

---

### Official Review · Reviewer_NgNT · 2023-07-15

**Soundness:** 3 good
**Presentation:** 4 excellent
**Contribution:** 4 excellent
**Rating:** 8
**Confidence:** 4

**Summary:**

This paper aims to understand how private information leaks from gradients in model training.
To this end, the authors propose to use influence analysis to analyze how gradient perturbations can affect the quality of samples reconstructed from private gradients.
Specifically, they first prove that under some assumptions, the privacy leaked from a perturbed gradient can be strictly characterized by a novel Inversion Influence Function (I2F), which is a function that takes the gradient perturbation as input.
With I2F, the authors then gain insights about DGL, such as (1) common random noise (e.g., Gaussian noise) can not provide uniformly fair privacy protection for each gradient, (2) future protection for DGL can be developed based on the eigenvalues of the Jacobian matrices of the model.

**Strengths:**

This paper makes the first step to address an important and fundamental problem, i.e., decoupling the black-box behavior of data gradient leakage. The authors make non-trivial contributions in various aspects:

1. The authors design a novel influence function named I2F based on non-trivial influence analysis. It can strictly characterize the privacy protection provided by gradient perturbation, which gives the first theoretical result about how to analyze gradient privacy leakage.

2. Beyond theoretical analysis, the authors also conduct empirical studies to justify the feasibility of using I2F to study DGL in practice.

3. Based on I2F, the authors found that the singular values of model Jacobian with respect to single inputs could be a good indicator for representing privacy protection strengths, which then illustrates that the gradient privacy protection provided by random noise (which is commonly used in practice) could be unfair to different training data.

4. The paper first establishes the connection between model Jacobian and gradient privacy protection, which suggests that future defense mechanisms can be designed starting from the singular values of model Jacobian.

**Weaknesses:**

1. The main weakness is that the experiments could be not comprehensive enough to justify the effectiveness of I2F. Specifically:
    - Only two datasets, MNIST and CIFAR-10, are involved in the experiments. Furthermore, one of the datasets MNIST could be too simple for illustrating representative results.

    - Only a few models are used in the experiments and most of them could be too small.

2. Suggestion: I think it is not necessary to use $\mathcal I_{\mathrm{lb}}$ as a replacement to approximate I2F since the calculation of I2F seems not really costly. Specifically, the main calculation cost would be calculating the matrix inversion of $JJ^T$. Because $JJ^T$ is (assumed to be) positive definite, you can easily use `numpy.linalg.eigh()` to calculate the matrix inverse, which is usually calculation efficient in practice.

======== After Rebuttal ========

Score has been raised based on the authors response.

**Questions:**

None.

**Limitations:**

This paper contributes non-trivially to understanding how privacy leaks from gradients.
Although the empirical analysis could be a little weak, I think this work will significantly contribute to the ML security community and therefore should be accepted.

---

> ### Author Rebuttal · Authors · 2023-08-10
>
> We really appreciate the affirmation from the reviewer of our contributions and innovations. We are glad to address the concerns as follows:
>
> **W1:** *Only MNIST and CIFAR10 are in experiments. MNIST is a little simple. Also, the models are too small*
>
> **A1**: Thanks for the suggestion. In Fig.5 of the attached PDF, we evaluate our metric on the large model (ResNet152) and large dataset (ImageNet).
> 1. For larger models, the MSE is no longer a good metric for the recovery evaluation. Even if state-of-the-art attacks are used and the recovered image is visually similar to the original image in Fig 2(b), the two images are measured to be different by MSE, due to the visual shift: The dog head is shifted toward the left side. To capture such shifted similarity, we use LPIPS [1] instead, which measures the semantic distance between two images instead of the pixel-to-pixel distance like MSE.
> 2. Fig.2(a) shows that I2F is correlated to LPIPS using large models and image scales. This implies that **I2F is a good estimator of recovery similarity even for large models and images**.
>
> **W2:** *No need to approximate I2F with I_{lb}, since we can use numpy.linalg.eigh() to calculate matrix inverse.*
>
> **A2:** A: Thanks for the suggestion. It is a good point that we can use numpy.linalg.eigh() to calculate the matrix inverse. However, for high dimensions of parameters and input images, the main challenge involves the computation of the full Jacobian matrix $J$, besides the inverse. Thus, we propose two alternative approaches in “Efficient Evaluation” to efficiently compute $(JJ^T)^{-1} Jv$ where v is a vector, without explicitly obtaining $JJ^T$.
>
> [A] Zhang, Richard, et al. "The unreasonable effectiveness of deep features as a perceptual metric." CVPR. 2018.

---

> > ### Comment · Reviewer_NgNT · 2023-08-19
> > **Score has been raised**
> >
> > Thanks to the authors for their response. All of my questions have been resolved. I believe this work will significantly contribute to the ML security community so I have raised my score from 7 to 8 to strongly vote for acceptance.

---

> > > ### Author Response · Authors · 2023-08-19
> > > **Thanks for raising the score**
> > >
> > > We are glad that all your concerns are addressed. Many thanks for reading our responses and lettering us know your thoughts. We really appreciate your valuable comments and acknowledging the contribution of our work!

---

### Author Rebuttal · Authors · 2023-08-10

Thanks to all the reviewers for their patient reading and valuable comments. We are trying our best to address the concerns of all the reviewers. Here we attach a PDF for more empirical results.

---

### Decision · Program_Chairs · 2023-09-21

**Decision:**

Accept (poster)

**Comment:**

This paper proposes a novel influence function-based analysis of gradient perturbation. Using this analysis, the authors show that top singular vectors of the gradient Jacobian matrix can predict the optimal direction of perturbation to protect against gradient inversion, and reveals interesting insights such that privacy protection is not equal across training samples. Reviewers unanimously agree that the analysis is novel and can be used to aid the design of future inversion attacks and defense mechanisms.

The most major weakness is lack of discussion about prior work. Reviewer BBkr suggested two relevant works on bounding data reconstruction attacks against DP-SGD (Guo et al. https://arxiv.org/abs/2201.12383 and Hayes et al. https://arxiv.org/abs/2302.07225). In fact, gradient inversion is a special case of data reconstruction when the model is trained for a single step from a known intermediate state. Moreover, the analysis of Guo et al. using Fisher information, when specialized to the single step setting, is also based on the gradient Jacobian matrix and should be more thoroughly discussed and compared against.

Ultimately, reviewers agree that the authors' analysis is sufficiently different from prior work to be considered novel, but urge the authors to more carefully position their work in light of prior works, especially that of Guo et al. AC recommends acceptance conditioned on this modification to the manuscript.